# STAMP: Scalable Task- And Model-agnostic Collaborative Perception

**Xiangbo Gao[1], Runsheng Xu[2], Jiachen Li[3], Ziran Wang[4], Zhiwen Fan[5], Zhengzhong Tu[1]***
[1]Texas A&M University, [2]UCLA, [3]UC Riverside, [4]Purdue University, [5]UT Austin
{xiangbog,tzz}@tamu.edu

## Abstract

Perception is a crucial component of autonomous driving systems. However, single-agent setups often face limitations due to sensor constraints, especially under challenging conditions like severe occlusion, adverse weather, and long-range object detection. Multi-agent collaborative perception (CP) offers a promising solution that enables communication and information sharing between connected vehicles. Yet, the heterogeneity among agents—in terms of sensors, models, and tasks—significantly hinders effective and efficient cross-agent collaboration. To address these challenges, we propose STAMP, a scalable task- and model-agnostic collaborative perception framework tailored for heterogeneous agents. STAMP utilizes lightweight adapter-reverter pairs to transform Bird's Eye View (BEV) features between agent-specific domains and a shared protocol domain, facilitating efficient feature sharing and fusion while minimizing computational overhead. Moreover, our approach enhances scalability, preserves model security, and accommodates a diverse range of agents. Extensive experiments on both simulated (OPV2V) and real-world (V2V4Real) datasets demonstrate that STAMP achieves comparable or superior accuracy to state-of-the-art models with significantly reduced computational costs. As the first-of-its-kind task- and model-agnostic collaborative perception framework, STAMP aims to advance research in scalable and secure mobility systems, bringing us closer to Level 5 autonomy. Our project page is at https://xiangbogaobarry.github.io/STAMP and the code is available at https://github.com/taco-group/STAMP.

## 1 Introduction

Multi-agent collaborative perception (CP) (Bai et al., 2022b; Han et al., 2023; Liu et al., 2023a) has emerged as a promising solution for autonomous systems by leveraging communication among multiple connected and automated agents. It enables agents—such as vehicles, infrastructure, or even pedestrians—to share sensory and perceptual information, providing a more comprehensive view of the surrounding environment to enhance overall perception capabilities. Despite its potential, CP faces significant challenges, particularly when dealing with heterogeneous agents that defer in input modalities, model parameters, architectures, or learning objectives. For instance, Xu et al. (2023b) observed that features from heterogeneous agents vary in spatial resolution, channel number, and feature patterns. This domain gap hinders effective and efficient CP, particularly when employing fusion-based approaches.

To facilitate collaborative perception among heterogeneous agents—often referred to as heterogeneous collaborative perception—one might consider using early or late fusion. However, early fusion requires high communication bandwidth, making it impractical for real-time applications. Late fusion often results in suboptimal accuracy, and it is not viable across models with different downstream tasks. Alternative methods attempt to achieve heterogeneous intermediate fusion by either incorporating adapters (Xu et al., 2023b) or sharing parts of the models (Lu et al., 2024). While these approaches can bridge the domain gap, they are limited in scalability or security, rendering them inefficient or unsafe for practical deployment. Additionally, recent studies have highlighted increased security vulnerabilities in CP systems compared to single-agent frameworks (Hu et al.,

---
*Corresponding author.

2024; Tu et al., 2021; Li et al., 2023b). Notably, Li et al. (2023b) found that black-box attacks are nearly ineffective without the knowledge of other agents' models, emphasizing the importance of task- and model-agnostic approaches in enhancing system-level security against adversarial threats.

To address these challenges, we propose **STAMP**, a Scalable Task- And Model-agnostic collaborative Perception framework. Our approach employs lightweight adapter-reverter pairs to transform the Bird's Eye View (BEV) features of each heterogeneous agent into a unified protocol BEV feature domain. These protocol BEV features are then broadcasted to other agents and subsequently mapped back to their corresponding local domains, enabling collaboration within each agent's source domain. We refer to this process as **collaborative feature alignment (CFA)**. Our proposed pipeline offers several key advantages. Firstly, it enables existing heterogeneous agents to collaborate with minimal additional disk memory ($\sim$1MB) and computational overhead, making it scalable for a large number of heterogeneous agents. Secondly, the alignment process is designed to be task- and model-agnostic, allowing our framework to integrate with various models and tasks without retraining the model or the need to share models among agents, enhancing both *flexibility* and *security*.

We conducted comprehensive experiments to evaluate the performance of our collaborative perception framework. Using the simulated OPV2V dataset (Xu et al., 2022b) and the real-world V2V4Real dataset (Xu et al., 2023d), we demonstrated that our STAMP pipeline achieves comparable or superior accuracy with a significantly lower training resource growth rate as the number of heterogeneous agents increases. Our method requires, on average, only 2.36 GPU hours (**7.2x** saving) of training time per additional agent, compared to 17.07 GPU hours per additional agent for existing heterogeneous collaborative pipelines. We also demonstrated our pipeline's unique ability to perform *task- and model-agnostic* collaboration, a capability *not supported in existing methods*. This achievement establishes a new benchmark for heterogeneous CP in autonomous driving, showcasing clear performance improvements in scenarios where other methods are unable to operate.

## 2 RELATED WORKS

### 2.1 MULTI-AGENT COLLABORATIVE PERCEPTION

Multi-agent CP has emerged as a promising solution to overcome the inherent limitations of single-agent perception systems, particularly in addressing occlusions and extending perception range (Hu et al., 2022a).

**Information-sharing schemes.** There are three main information-sharing schemes in multi-agent CP systems: early fusion, late fusion, and intermediate fusion. ❶ Early fusion (Gao et al., 2018; Chen et al., 2019; Arnold et al., 2020) involves the direct sharing of raw sensor data, such as LiDAR point clouds or camera images, between agents. This method maximizes information transfer but requires high bandwidth for transmitting. ❷ Late fusion (Melotti et al., 2020; Fu et al., 2020; Zeng et al., 2020; Shi et al., 2022; Glaser & Kira, 2023; Xu et al., 2023a; Su et al., 2023; 2024), involves sharing only final prediction results, such as object detection bounding boxes or occupancy predictions. This approach significantly reduces communication bandwidth overhead, making it more feasible for implementation in real-world systems. However, late fusion often results in suboptimal accuracy due to the loss of intermediate information that could be valuable for collaborative decision-making. ❸ Intermediate fusion (Wang et al., 2020; Liu et al., 2020b;a; Guo et al., 2021; Li et al., 2021; Hu et al., 2022b; Bai et al., 2022a; Cui et al., 2022; Xu et al., 2022a; 2023c; Qiao & Zulkernine, 2023; Li et al., 2023a; Wang et al., 2023; Yu et al., 2023; Yang et al., 2024) has emerged as a promising middle ground, involving the sharing of mid-level information, typically in the form of Bird's Eye View (BEV) features. This approach strikes a balance between communication bandwidth efficiency and information richness. Intermediate fusion allows for more flexibility in collaborative processing while maintaining a reasonable data transfer load. However, intermediate fusion faces significant challenges in addressing domain gap issues for heterogeneous agents.

**Collaborative perception datasets.** Several significant collaborative perception datasets have emerged recently (Yazgan et al., 2024). The simulated OPV2V (Xu et al., 2022b) and V2X-Sim dataset (Li et al., 2022) each contain approximately $10k$ multi-agent scenes, featuring RGB images and LiDAR point clouds with 3D object detection, tracking, and segmentation annotations. Two real-world DAIR-V2X Yu et al. (2022) and V2V4Real (Xu et al., 2023d) datasets provide $39k$ and $20k$ dual-agent samples, respectively, with object detection annotations only. For our framework

evaluation, we selected two complementary datasets: the OPV2V dataset for its diverse downstream tasks, and the V2V4Real dataset to validate performance in real-world scenarios.

## 2.2 HETEROGENEOUS COLLABORATIVE PERCEPTION

In a CP system, the heterogeneity of agents can manifest as three different types: heterogeneous modalities, heterogeneous model architectures or parameters, and heterogeneous downstream tasks. ❶ **Heterogeneous modalities.** Each model is expected to take input data of different modalities (e.g., RGB images, LiDAR point clouds (Liu et al., 2023b), Thermal images (Gao et al., 2024a)), requiring different encoders to process the data. Xiang et al. (2023) propose a hetero-modal vision transformer to fuse heterogeneous BEV features, but this requires end-to-end model training, which is impractical for existing heterogeneous agents. Xu et al. (2023b) introduce multi-agent perception domain adaptation (MPDA), which aligns feature maps between heterogeneous agent pairs. While effective for collaboration, this method's polynomial complexity limits its scalability as the number of heterogeneous models increases. Lu et al. (2024) introduce a backward alignment training strategy, creating heterogeneous models by fixing a base network's decoder and training only the encoders. While this enables collaboration between existing heterogeneous agents, it incurs high computational costs, especially for models with large encoders. ❷ **Heterogeneous model architectures or parameters.** Model architectures or parameters may differ across agents, resulting in feature map in different domains, rendering existing heterogeneous intermediate fusion methods (Xiang et al., 2023; Lu et al., 2024) inapplicable. However, late fusion methods (Xu et al., 2023b) remain viable as the model output for all models is in the same domain. ❸ **Heterogeneous downstream tasks.** The learning objectives are different across agents, which results in model outputs in different domains. Li et al. (2023c) propose task-agnostic CP by training models with multi-robot scene completion objectives. Despite the effectiveness of task-agnostic collaboration, their method does not support heterogeneous modality inputs and model architectures.

## 3 METHODOLOGY

### 3.1 PRELIMINARIES: INTERMEDIATE COLLABORATIVE PERCEPTION

A CP system typically comprises multiple ($N$) agents, each equipped with its own CP model. This work mainly focuses on intermediate fusion, so we consider all CP models to be trained using an intermediate fusion strategy. The architecture of these models generally consists of an encoder $E_i$, a compressor $\phi_i$, a decompressor $\psi_i$, a collaborative fusion layer $U_i$, and a decoder $D_i$, where $i \in \{1, 2, \ldots, N\}$ represents the agent index.

The CP process unfolds as follows: Upon receiving input data $I_i$, the encoder $E_i$ of agent $i$ transforms this data into a Bird's Eye View (BEV) feature representation $F_i$. To save transmission bandwidth, each agent uses the compressor $\phi_i$ to compress $F_i$ to $\tilde{F}_i$ before broadcasting them to other agents within a predefined collaborative distance $\delta$. Here, $\delta$ denotes the maximum range for inter-agent collaboration. Each agent collects the BEV features from other agents and uses their own decompressor $\psi_i$ to decompress $\tilde{F}_k$ to $F_k$, where agent $i$ and agent $k$ are within the distance $\delta$ for collaboration. Then, the collaborative fusion layer $U_i$ collects and integrates the BEV features from all cooperating agents, producing a consolidated BEV feature $F'$. Finally, the decoder $D_i$ processes this fused feature $F'$ to generate the final model output $O_i$. This process can be formally described as follows for each agent $i \in \{1, 2, \ldots, N\}$:

$$\text{Encoding:} \quad F_i = E_i(I_i) \tag{1}$$
$$\text{Compression:} \quad \tilde{F}_i = \phi_i(F_i) \tag{2}$$
$$\text{Decompression:} \quad F_j = \psi_i(\tilde{F}_j), \quad \forall j \in \mathcal{N}(i, j) \leq \delta \tag{3}$$
$$\text{Fusion:} \quad F_i' = U_i(\{F_j \mid \mathcal{N}(i, j) \leq \delta\}) \tag{4}$$
$$\text{Decoding:} \quad O_i = D_i(F_i') \tag{5}$$

where $\mathcal{N}(i, k)$ refers to the Euclidean distance between the agent $i$ and agent $k$.

### 3.2 FRAMEWORK OVERVIEW

Our proposed framework, STAMP, enables collaboration among existing heterogeneous agents without sharing model details or downstream task information. We replace the compression (Equation 2)

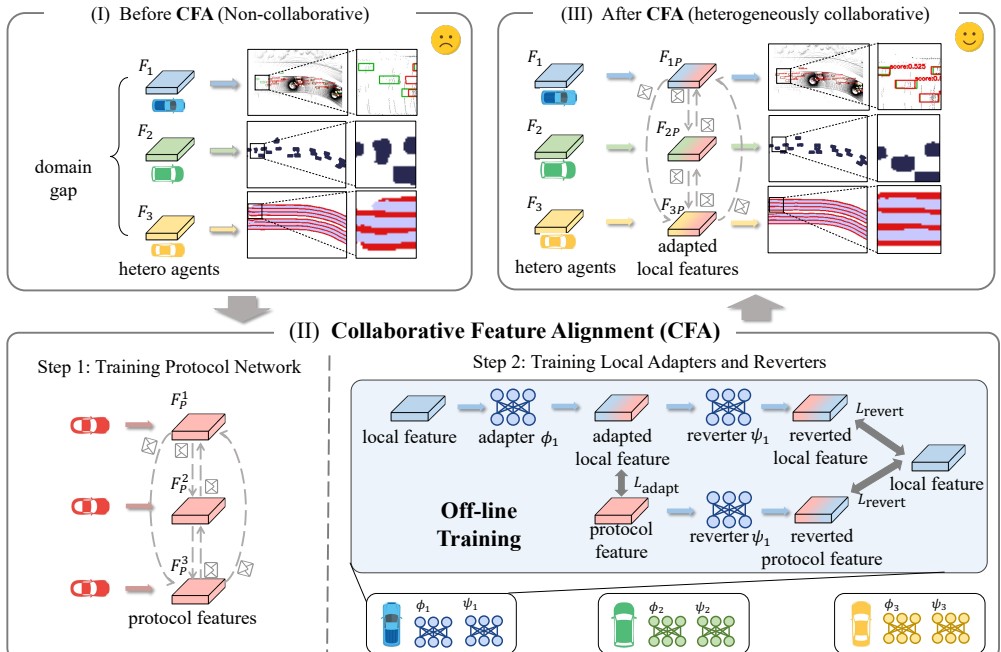

Figure 1: Initially, agents are non-collaborative (I), resulting in degraded performance. Collaborative Feature Alignment (CFA) enables collaboration among heterogeneous agents through a two-step process (II): training a protocol network and training local adapters and reverters. The protocol network facilitates communication between Agent 1, Agent 2, and Agent 3, each with heterogeneous models and features. Gradient-colored feature maps represent features adapted or reverted between domains. After CFA implementation, agents become collaborative (III) with improved performance.

and decompression (Equation 3) with adaptation and reversion steps. Specifically, for each agent $i$, we introduce a local adapter $\phi_i$ and a local reverter $\psi_i$. The adaptation process is defined as follows:

$$\text{Adaptation:} \quad F_{iP} = \phi_i(F_i), \quad \forall i \in \{1, 2, \ldots, N\} \tag{6}$$

Here, the adapter $\phi_i$ maps the local BEV feature $F_i$ to a unified BEV feature representation, which we term the protocol feature, denoted as $F_P$. The resulting adapted feature is denoted as $F_{iP}$.

Following adaptation, the features from all heterogeneous agents $i$ are broadcast to other agents $j$ within the collaborative distance $\delta$. Each receiving agent $j$ ($j \neq i$) then uses its local reverter $\psi_j$ to map the received features back to its own local feature representation. The resulting reverted features are denoted as $F_{ij}$. This reversion process is formulated as:

$$\text{Reversion:} \quad F_{ij} = \begin{cases} \psi_j(F_{iP}), & \text{if } i \neq j \\ F_i, & \text{if } i = j \end{cases} \quad \forall i, j \in \{1, 2, \ldots, N\} \tag{7}$$

Note that $F_i$ is already in the local feature presentation, so stay intact. This adaptation and reversion process seamlessly integrates into the standard heterogeneous CP pipeline, forming the core of our STAMP framework.

The STAMP framework supports agents with different modalities, model architectures, and downstream tasks, while maintaining a collaboration process that is entirely agnostic to those characteristics of other agents. To the best of our knowledge, STAMP is the first framework that simultaneously addresses all three aspects of agent heterogeneity. Furthermore, the adapters and reverters can be implemented in a highly lightweight manner, ensuring high scalability across a large number of heterogeneous agents. A comprehensive comparative summary STAMP and other heterogeneous CP frameworks is presented in Table 1.

## 3.3 COLLABORATIVE FEATURE ALIGNMENT

We propose the **Collaborative Feature Alignment (CFA)** module to train a unified BEV feature representation and a local adapter-reverter pair. As illustrated in Figure 1 (I), before CFA, het-

Table 1: Comparison of heterogeneity support and scalability across existing heterogeneous collaboration frameworks. "†" indicates that while the current codebase does not support the specified heterogeneity, we believe the proposed method could accommodate it with minor modifications.

| Frameworks | Modality | Model Architecture | Downstream Task | Scalability |
|---|---|---|---|---|
| Calibrator (Xu et al., 2023a) | ✓ | ✓ | | high |
| MPDA (Xu et al., 2023b) | ✓ | † | † | low |
| HEAL (Lu et al., 2024) | ✓ | | | medium |
| Scene Completion (Li et al., 2023c) | † | | ✓ | high |
| STAMP (**ours**) | ✓ | ✓ | ✓ | high |

erogeneous agents perform multiple tasks individually without the ability to collaborate, resulting in suboptimal performance. After CFA (III), these agents can effectively collaborate, leading to a significant performance boost.

❶ **Training protocol network.** The first step is to learn a unified BEV embedding space by training a protocol network. This process follows the standard training process of a collaborative perception model, as described in Equations (1) to (5). We denote the protocol encoder, fusion model, and decoder as $E_P$, $U_P$, and $D_P$, respectively. The compressor and decompressor are set as identity functions. The input data, BEV feature, fused BEV feature, and final output of the protocol model are represented by $I_P$, $F_P$, $F'_P$, and $O_P$, respectively.

❷ **Training local adapters and reverters.** We introduce a notation $\mathcal{X} = \{\mathbf{x}^1, \mathbf{x}^2, \ldots, \mathbf{x}^K\}$ to denote a set of **world states**, where $K$ is the total number of world states. A modality transformation function $\mathcal{T} : \mathcal{X} \to I$ transforms a world state to sensor data of a given modality. For instance, in an autonomous driving scenario, $\mathbf{x}$ could represent the world state surrounding the ego vehicle, and $\mathcal{T}$ could be the matrix of six surround-view RGB cameras, resulting in $I$ as the six RGB images captured by these cameras.

Given a local model $i$, protocol model $P$, and a set of world states $\mathcal{X} = \{\mathbf{x}^1, \mathbf{x}^2, \ldots, \mathbf{x}^K\}$ within the collaborative distance, we define $I_i^k = \mathcal{T}_i(\mathbf{x}^k)$ as the input for local model $i$ and $I_P^k = \mathcal{T}_P(\mathbf{x}^k)$ as the input for protocol model $P$. Here, $\mathcal{T}_i$ and $\mathcal{T}_P$ denote the sensor modalities of the local model $i$ and protocol model, respectively. By passing these inputs into their corresponding encoders, we get a set of local BEV features $F_i^{1:K}$ and protocol BEV features $F_P^{1:K}$. A domain gap exists between $F_i^{1:K}$ and $F_P^{1:K}$ due to the heterogeneity of sensor modalities $\mathcal{T}_i$ and $\mathcal{T}_P$, as well as the encoders $E_i$ and $E_P$. To bridge this gap, we introduce a local adapter $\phi_i$ that maps the local BEV feature $F_i^k$ to the protocol feature representation. The objective function for $\phi_i$ is:

$$\phi_i = \underset{\phi_i}{\arg\min}\, L_{\phi_i}(F_{iP}^{1:K}, F_P^{1:K}) \quad \text{where} \quad F_{iP}^k = \phi_i(F_i^k) \tag{8}$$

where $L_{\phi_i}$ represent the feature alignment loss for the adapter $\phi_i$. Similarly, we introduce a reverter $\psi_i$ that maps the protocol BEV feature $F_P^k$ to the local feature representation, *i.e.*, $F_{Pi}^k = \psi_i(F_P^k)$. since $F_{iP}^k$ is also in the protocol representation, by including $F_{ii}^k = \psi_i(F_{iP}^k)$, we provide additional supervision for $\psi_i$. The objective function for the reverter is formulated as:

$$\psi_i = \underset{\psi_i}{\arg\min}\, \left( L_{\psi_i}(F_{Pi}^{1:K}, F_i^{1:K}) + L_{\psi_i}(F_{ii}^{1:K}, F_i^{1:K}) \right)$$
$$\text{where} \quad F_{Pi}^k = \psi_i(F_P^k),\ F_{ii}^k = \psi_i(F_{iP}^k) \tag{9}$$

Here, $L_{\psi_i}$ represents the feature alignment loss for the reverter $\psi_i$. To achieve our objective functions, we conduct alignment in both the feature space and the decision space.

❸ **Feature space alignment.** For a given local model $i$, we first align the feature pairs $(F_{iP}^{1:K}, F_P^{1:K})$, $(F_{Pi}^{1:K}, F_i^{1:K})$, and $(F_{ii}^{1:K}, F_i^{1:K})$ for all $k$ using the $L2$-norm. This direct alignment of feature spaces is formulated as:

$$L_{\phi_i}^f = \frac{1}{K} \sum_k \|F_{iP}^k, F_P^k\|_2, \quad L_{\psi_i}^f = \frac{1}{K} \sum_k \left( \|F_{Pi}^k, F_i^k\|_2 + \|F_{ii}^k, F_i^k\|_2 \right) \tag{10}$$

❹ **Decision space alignment.** When $\mathcal{T}_i$ and $\mathcal{T}_P$ represent significantly different sensor modalities (e.g., RGB camera vs. LiDAR), the disparity between their intermediate feature representations can

be substantial. In such cases, achieving exact equivalence between $F_{iP}^k$ and $F_P^k$ for all $k$ is neither feasible nor necessary. Nevertheless, since both $F_i^k$ and $F_{iP}^k$ are derived from the same world state $\mathbf{x}^k$, their corresponding downstream task outputs should align with the same ground truth labels. To enforce this alignment in the decision space, we introduce additional loss terms:

$$
\begin{aligned}
L_{\phi_i}^d &= \mathcal{L}_P(D_P \circ U_P(F_{iP}^{1:K}),\ \mathrm{GT}_P) \\
L_{\psi_i}^d &= \mathcal{L}_i(D_i \circ U_i(F_{Pi}^{1:K}),\ \mathrm{GT}_i) + \mathcal{L}_i(D_i \circ U_i(F_{ii}^{1:K}),\ \mathrm{GT}_i)
\end{aligned} \tag{11}
$$

where $\mathcal{L}_P$ and $\mathcal{L}_i$ represent the task-specific loss functions for training the protocol model and local model $i$, respectively. $\mathrm{GT}_P$ and $\mathrm{GT}_i$ denote the corresponding ground truth labels for the protocol and local model $i$. Finally, to balance the importance of adaptation and reversion, as well as feature and decision space alignment, we introduce scaling factors $\lambda_\phi^f$, $\lambda_\psi^f$, $\lambda_\phi^d$, and $\lambda_\psi^d$. The total loss function for local model $i$ is:

$$
L_i = \lambda_\phi^f L_{\phi_i}^f + \lambda_\psi^f L_{\psi_i}^f + \lambda_\phi^d L_{\phi_i}^d + \lambda_\psi^d L_{\psi_i}^d \tag{12}
$$

### 3.4 Adapter and Reverter Architecture

To bridge the domain gap between heterogeneous agents, we propose a flexible architecture for both the adapter $\phi$ and reverter $\psi$. This architecture addresses three main sources of domain gap caused by agent heterogeneity, as identified by Xu et al. (2023b): spatial resolution, feature patterns, and channel dimensions. Our design employs simple linear interpolation for spatial resolution alignment, three ConvNeXt layers (Liu et al., 2022) with hidden channel dimension $C_{\text{hidden}}$ for feature pattern alignment, and two additional convolutional layers for channel dimension alignment (input: $C_{\text{in}} \rightarrow C_{\text{hidden}}$, output: $C_{\text{hidden}} \rightarrow C_{\text{out}}$). For the model architecture details, please refer to Appendix A.1.

Note that this high-level architecture is flexible and open to various implementations. In Section 4.4, we evaluate alternative approaches for feature pattern alignment, demonstrating our framework's flexibility across different specific implementations.

## 4 Experiments

Our STAMP framework enables collaboration among agents with heterogeneous modalities, models architectures, and downstream tasks without sharing model or task information. We first compare our framework with existing heterogeneous CP frameworks in Section 4.2. Given that no previous work supports simultaneous task- and model-agnostic heterogeneous collaboration, we concentrate our evaluation on the 3D object detection task. This focus ensures a fair comparison across two key dimensions: object detection average precision, trainable parameters and GPU hours required for training. Next, in Section 4.3, we demonstrate our framework's unique capability in a task- and model-agnostic setting, evaluating its performance using four existing collaborative models with heterogeneous architectures and downstream tasks. Then, we present ablation studies on channel sizes, model architectures, and loss functions in Section 4.4 to further analyze our framework's design choices. Finally, we present some feature and output visualization in Section 4.5.

### 4.1 Experimental Setup

Our experiments utilize two CP datasets: the simulated OPV2V dataset (Xu et al., 2022b) and the real-world V2V4Real dataset (Xu et al., 2023d). We employ both datasets in Section 4.2 and Section 4.4 for method comparison and ablation studies. The task- and model-agnostic evaluation in Section 4.3 uses only OPV2V due to its multi-task annotations. This combination leverages the scale and diversity of simulated data with the realism of real-world data, ensuring comprehensive model evaluation.

**Implementation details.** We use different setups for 3D object detection (Section 4.2) and task-agnostic settings (Section 4.3), detailed within each section. Unless using end-to-end training, local and protocol models are trained for 30 epochs using Adam optimizer (Kingma & Ba, 2014). For end-to-end training, we use $\text{Iters}_N = 30N$ epochs, where $N$ is the number of heterogeneous models, to ensure all models receive the same amount of supervision. Local adapters $\phi$ and reverters $\psi$ are trained for 5 epochs. We set loss scaling factors $\lambda_{\text{adapt}}^f = \lambda_{\text{revert}}^f = \lambda_{\text{adapt}}^d = \lambda_{\text{revert}}^d = 0.5$ empirically. For additional details, please refer to Section 4.2, Section 4.3, and Appendix A.1.

Table 2: Performance comparison using AP@30 and AP@50 metrics on the OPV2V dataset. Agent positions are perturbed with Gaussian noise of standard deviations 0.0, 0.2, and 0.4. A1, A2, A3, and A4 refer to agent 1, agent 2, agent 3, and agent 4, respectively.

| $\sigma$ | Agent Index | AP@30 ↑ | | | | AP@50 ↑ | | | |
|---|---|---|---|---|---|---|---|---|---|
| | | **A1** | **+A2** | **+A3** | **+A4** | **A1** | **+A2** | **+A3** | **+A4** |
| 0.0 | Late Fusion | **0.902** | 0.931 | 0.935 | 0.935 | 0.894 | 0.908 | 0.913 | 0.914 |
| | Calibrator | 0.901 | 0.935 | 0.939 | 0.938 | **0.896** | 0.914 | 0.916 | 0.920 |
| | E2E Training | 0.899 | 0.973 | 0.978 | 0.986 | 0.885 | 0.967 | 0.977 | 0.980 |
| | HEAL | 0.897 | 0.975 | 0.986 | 0.985 | 0.889 | 0.972 | 0.978 | 0.983 |
| | STAMP (ours) | **0.902** | **0.981** | **0.987** | **0.989** | 0.894 | **0.977** | **0.983** | **0.985** |
| 0.2 | Late Fusion | **0.900** | 0.910 | 0.902 | 0.905 | 0.882 | 0.783 | 0.797 | 0.792 |
| | Calibrator | 0.897 | 0.908 | 0.898 | 0.902 | **0.885** | 0.778 | 0.800 | 0.791 |
| | E2E Training | **0.900** | 0.967 | 0.961 | 0.961 | 0.879 | 0.936 | 0.941 | 0.934 |
| | HEAL | 0.899 | 0.971 | 0.965 | 0.962 | 0.881 | **0.938** | 0.940 | 0.940 |
| | STAMP (ours) | **0.900** | **0.975** | **0.968** | **0.968** | 0.882 | 0.937 | **0.948** | **0.946** |
| 0.4 | Late Fusion | **0.888** | 0.882 | 0.864 | 0.870 | **0.874** | 0.639 | 0.614 | 0.617 |
| | Calibrator | 0.874 | 0.889 | 0.887 | 0.871 | 0.867 | 0.644 | 0.622 | 0.628 |
| | E2E Training | 0.885 | 0.952 | 0.956 | 0.952 | 0.863 | 0.883 | 0.892 | 0.899 |
| | HEAL | 0.880 | 0.959 | 0.961 | 0.962 | 0.852 | 0.913 | **0.915** | **0.912** |
| | STAMP (ours) | **0.888** | **0.961** | **0.963** | **0.966** | **0.874** | **0.915** | **0.915** | 0.909 |

## 4.2 HETEROGENEOUS COLLABORATIVE PERCEPTION FOR 3D OBJECT DETECTION

**Performance comparison.** We compare our method with existing heterogeneous CP approaches on the 3D object detection task. We select two late fusion methods (vanilla late fusion and calibrator (Xu et al., 2023a)) and two intermediate fusion methods (end-to-end training and HEAL (Lu et al., 2024)) for comparison. Late fusion methods offer a simple way to mitigate domain gaps in collaborative 3D object detection. Xu et al. (2023a) propose using a calibrator to address residual domain gaps in late fusion, which arise from differences in training data and procedures among heterogeneous models. For intermediate fusion, end-to-end training of all heterogeneous models together allows collaboration during the training stage to bridge domain gaps. Lu et al. (2024) introduce a backward alignment technique, first training a base network, then fixing its decoder while training only the encoders to create heterogeneous models. An architectural comparison between these frameworks and our proposed STAMP framework is illustrated and visualized in Appendix A2.

We prepared 12 heterogeneous local models (six with LiDAR modality and six with RGB camera modality) and one protocol model with LiDAR modality (details in Appendix A.1). Each agent has a visible range of $51.2\text{m} \times 51.2\text{m}$ square units. Considering that most samples of the OPV2V dataset contain no more than four agents, we only select the first four models for evaluation on the OPV2V dataset. Similarly, we select the first two models for the V2V4Real dataset since it has two agents for each sample. All 12 models are used for efficiency comparison.

For the OPV2V dataset, we simulate real-world noise by adding Gaussian noise with standard deviations $\sigma = \{0.0, 0.2, 0.4\}$ to the agents' locations. As shown in Table 2, late fusion methods underperform as the number of agents increases, with performance degrading further at higher noise levels ($\sigma = 0.4$). This is particularly evident when camera agents (agents 3 and 4) are involved, highlighting the late fusion methods' vulnerability to bottleneck agents' incorrect predictions. Our framework demonstrates superior or comparable performance to other heterogeneous fusion methods across all noise levels.

Table 3 compares the average precision on the real-world V2V4Real dataset. Our STAMP pipeline demonstrates superior performance, achieving the highest AP@30 for both agents (0.523 and 0.633) and competitive AP@50 scores. STAMP outperforms Late Fusion methods and matches or exceeds the performance of existing heterogeneous intermediate fusion approaches like HEAL. These results indicate that the CFA module in STAMP is effective not only in simulated environments but also in real-world scenarios.

Table 3: Performance comparison using AP@30 and AP@50 metrics on the V2V4Real dataset.

| Agent Index | AP@30 ↑ | | AP@50 ↑ | |
|---|---|---|---|---|
| | **A1** | **+A2** | **A1** | **+A2** |
| Late Fusion | 0.523 | 0.511 | 0.483 | 0.471 |
| Calibrator | 0.520 | 0.524 | 0.484 | 0.488 |
| E2E Training | 0.513 | 0.612 | 0.473 | 0.598 |
| HEAL | 0.515 | 0.628 | 0.480 | **0.595** |
| STAMP (ours) | **0.523** | **0.633** | **0.483** | 0.594 |

**Efficiency comparison.** We conducted an efficiency comparison between our approach and existing heterogeneous CP pipelines, focusing on the total number of parameters and training GPU hours. Training GPU hours refers to the time required to complete model training on the OPV2V dataset using an RTA A6000 GPU. To analyze training costs at scale, we report the number of training parameters and the estimated training time. Figure 2 illustrates the changes in the number of training parameters and estimated training

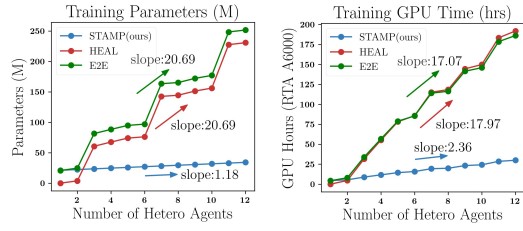

Figure 2: Training efficiency comparison of our framework and existing heterogeneous CP frameworks across a number of heterogeneous agents.

GPU hours as the number of heterogeneous agents increases from 1 to 12. End-to-end training and HEAL exhibit a steep increase in both parameters and GPU hours as the number of agents grows. In contrast, although our pipeline shows higher parameters and GPU hours at the one or two number of agents (due to the training of the protocol model), it demonstrates a much slower growth rate because our proposed adapter $\phi$ and reverter $\psi$ is very light-weighted and only takes 5 epochs to finish training. This highlights the scalability of our pipeline.

## 4.3 MODEL- AND TASK-AGNOSTIC FUSION

In this section, we evaluate our proposed framework's performance in a task-agnostic setting using the OPV2V dataset. To simulate agent heterogeneity, we assign four agents with diverse input sensors, learning objectives, and evaluation metrics, equipping them with various backbones and fusion models. Agent 1 was equipped with a SECOND encoder (Yan et al., 2018) and a window attention fusion module (Xu et al., 2022a; 2024). For Agent 2, we implemented an EfficientNet-b0 encoder (Tan, 2019), while Agents 3 and 4 were equipped with PointPillar encoders (Lang et al., 2019). Agents 2, 3, and 4 all utilized the Pyramid Fusion module Lu et al. (2024). Table 4 summarizes these models' key characteristics. We compare our STAMP framework against two baseline scenarios: non-collaborative (single-agent perception without information sharing) and collaborative without feature alignment (performing intermediate fusion despite domain gaps). Table 4 presents the evaluation results on the OPV2V dataset, with added Gaussian noise (standard deviations $\sigma = \{0.0, 0.2, 0.4\}$) to the agents' locations.

Table 4: Heterogeneous CP results in a model- and task-agnostic setting. Tasks include 3D object detection ('Object Det'), static object BEV segmentation ('Static Seg'), and dynamic object BEV segmentation ('Dynamic Seg'). 3D object detection is evaluated using Average Precision at 50% IoU threshold (AP@50), while segmentation tasks use Mean Intersection over Union (MIoU).

| | Agent Index | **Agent 1** | **Agent 2** | **Agent 3** | **Agent 4** |
|---|---|---|---|---|---|
| **Agent Info** | Metric
Downstream Task
Sensor Modality
Backbone
Feature Resolution
Channel Size
Fusion Method | AP@50
Object Det
Lidar
SECOND
$64 \times 64$
256
Window Attention | AP@50
Object Det
Camera
EfficientNet-b0
$128 \times 128$
64
Pyramid Fusion | MIoU
Static Seg
Lidar
PointPillar
$128 \times 128$
64
Pyramid Fusion | MIoU
Dynamic Seg
Lidar
PointPillar
$128 \times 128$
64
Pyramid Fusion |
| **Evaluation**
$(\sigma : 0.0)$ | Non-Collab
Collab w/o. CFA
STAMP (ours) | **0.941**
0.909 ↓0.032
0.936 ↓0.005 | 0.399
0.399→0.000
**0.760** ↑0.362 | 0.548
0.114 ↓0.434
**0.624** ↑0.076 | 0.675
0.070 ↓0.605
**0.690** ↑0.014 |
| **Evaluation**
$(\sigma : 0.2)$ | Non-Collab
Collab w/o. CFA
STAMP (ours) | **0.936**
0.902 ↓0.034
0.930 ↓0.006 | 0.399
0.399 →0.000
**0.734** ↑0.336 | 0.521
0.114 ↓0.407
**0.615** ↑0.094 | 0.658
0.069 ↓0.588
**0.676** ↑0.018 |
| **Evaluation**
$(\sigma : 0.4)$ | Non-Collab
Collab w/o. CFA
STAMP (ours) | **0.925**
0.886 ↓0.039
0.923 ↓0.002 | 0.399
0.400 ↑0.001
**0.585** ↑0.186 | 0.503
0.114 ↓0.389
**0.600** ↑0.097 | 0.630
0.069 ↓0.561
**0.650** ↓0.020 |

Our method consistently outperforms single-agent segmentation for agents 3 and 4 in the BEV segmentation task. Conversely, collaboration without feature alignment significantly degrades per-

formance compared to the single-agent baseline, underscoring the importance of our adaptation mechanism in aligning heterogeneous features. For agent 2's camera-based 3D object detection, our pipeline achieves substantial gains (e.g., AP@50 improves from 0.399 to 0.760 in noiseless conditions), while collaboration without feature alignment shows negligible changes. These results demonstrate our pipeline's effectiveness in bridging domain gaps between heterogeneous agents, enabling successful collaboration across diverse models, sensors, and tasks. The consistent improvements, particularly under noisy conditions, highlight our approach's robustness and adaptability.

However, we observe that both collaborative approaches lead to performance degradation for Agent 1 compared to its single-agent baseline, despite our method outperforming collaboration without feature alignment. This unexpected outcome is attributed to Agent 2's limitations, which rely solely on less accurate camera sensors for 3D object detection. This scenario illustrates a bottleneck effect, where a weaker agent constrains the overall system performance, negatively impacting even the strongest agents. This challenge in multi-agent collaboration systems prompts us to introduce the concept of a **Multi-group Collaboration System**. In Appendix A.4, we elaborate on the advantages of such a system and demonstrate how our framework can be easily integrated to potentially mitigate performance discrepancies in heterogeneous agent collaborations.

## 4.4 ABLATION STUDIES

In this section, we conduct ablation studies on three factors that may affect our pipeline's performance: BEV feature channel size, adapter & reverter architectures, and loss functions for collaborative feature alignment. All experiments are conducted on both the OPV2V and V2V4Real datasets.

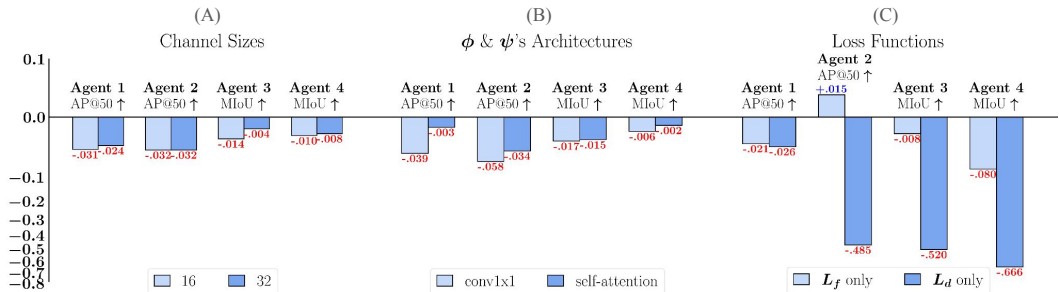

Figure 3: Ablation studies on the OPV2V dataset: (a) Model performance across different BEV feature channel sizes. (b) Performance comparison of various adapter and reverter architectures. (c) Performance results using different combinations of loss function components ($L_f$ and $L_d$).

**BEV feature channel size.** Changing the BEV feature channel size is essentially a form of feature compression, which is crucial for controlling communication bandwidth in multi-agent collaboration systems. Our collaborative feature alignment module inherently supports feature compression by adjusting the protocol BEV feature's channel size. We experiment with two channel sizes for the protocol BEV feature, 32 and 16, and compare their performance to our standard implementation with a channel size of 64 (Figures 3 and 4). Surprisingly, reducing the channel size results in only

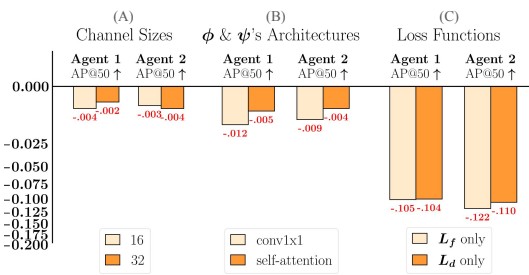

Figure 4: Ablation studies on the V2V4real set.

minor performance changes for both datasets, revealing our model's resilience to high BEV feature compression rates.

**Adapter & reverter architecture.** We evaluate two alternative architectures for the adapter and reverter—a single $1 \times 1$ convolutional layer and three self-attention layers—compared to our standard implementation of three ConvNeXt layers. The results demonstrate that performance is not highly

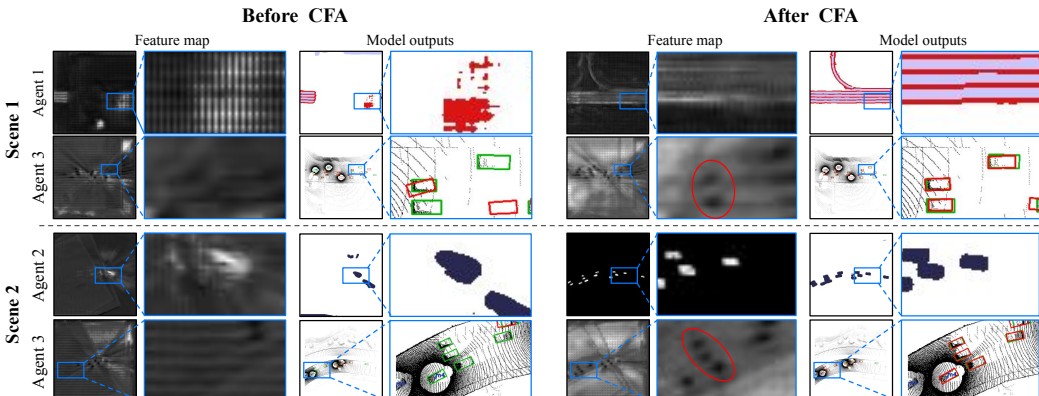

Figure 5: Visualization of feature maps and model outputs before and after Collaborative Feature Alignment (CFA) for two scenes with different agents and tasks. For 3D object detection, green boxes indicate the ground truth labels and red boxes indicate the predictions. CFA enhances feature clarity and information preservation, resulting in improved perception accuracy across heterogeneous agents.

sensitive to the adapter and reverter architecture, showcasing our framework's flexibility across various specific implementations.

**Loss function.** The final loss function in our collaborative feature alignment module comprises two components: $L_f$ for feature space alignment and $L_d$ for decision space alignment. We evaluate our method's performance using each loss function individually. Figure 3 shows that on the OPV2V dataset, using only $L_d$ leads to a significant performance drop, while using only $L_f$ results in more fluctuating and generally lower performance. Figure 4 demonstrates that on the V2V4Real dataset, dropping either $L_d$ or $L_f$ results in performance degradation. These findings underscore the necessity of using both loss functions in combination for optimal performance.

## 4.5 VISUALIZATION

Figure 5 illustrates the impact of our CFA method on feature maps and output results across various tasks. We visualize feature maps by averaging each channel of the fused feature map, $F_i'$, to a $(W, H)$ shape and plotting in a grayscale. Without CFA, the fused feature maps appear noisy and lack critical information for downstream tasks, leading to poor output results. In contrast, CFA significantly enhances feature preservation, resulting in clearer feature maps and more accurate outputs across different tasks. This visualization demonstrates CFA's effectiveness in maintaining essential information during the fusion process, which directly translates to improved performance in CP tasks. More comprehensive visualization results are shown on the Appendix A.5.

## 5 CONCLUSION

In this paper, we introduce STAMP, a scalable, task- and model-agnostic multi-agent collaborative perception framework. This framework simultaneously addresses three aspects of agent heterogeneity: varieties in modalities, model architectures, and downstream learning tasks. By utilizing lightweight adapter-reverter pairs, STAMP enables efficient collaborative perception while maintaining high security, scalability, and flexibility. Experiments on both the simulated OPV2V dataset and the real-world V2V4Real datasets demonstrate its superior performance and computational efficiency over existing state-of-the-arts. This approach opens new avenues for developing more reliable, efficient, and secure collaborative systems in future autonomous driving applications.

**Limitations.** Our experiments revealed a bottleneck effect in Collaborative Perception (CP), where the performance of the weakest agent constrains the overall system performance. This finding underscores the necessity for multi-group collaborative systems, where agents communicate only within defined groups. Such systems could mitigate the bottleneck effect by allowing for more selective collaboration. In Appendix A.4, we provide a more detailed discussion of multi-group collaborative systems and the advantages of our framework in this context.

**Reproducibility statement.** To ensure the reproducibility of our results, we have provided detailed information about our experimental setup, including dataset descriptions, model architectures, and training procedures in the main text and appendices. We encourage researchers to refer to Appendix A.1 for more implementation details. We also release the codebase at https://github.com/taco-group/STAMP.

**Ethics statement.** Our task- and model-agnostic framework enhances local model security, reducing risks like model stealing (Oliynyk et al., 2023) and adversarial attacks (Tu et al., 2021). While limiting model sharing improves security, as assessed by (Li et al., 2023b), we recognize the need for further security analysis. We advocate for collaboration with experts to rigorously evaluate and strengthen our approach in order to contribute to safer and more trustworthy autonomous driving systems and advance privacy in collaborative perception.

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

# A APPENDIX

## A.1 IMPLEMENTATION DETAILS

For training all models, we initialize the learning rate at $0.001$ and reduce it by a factor of $0.1$ at $50\%$ and $83\%$ of the total epochs. We utilize a single NVIDIA RTX A6000 GPU for both model training and inference. Training time for each model varies between 7 to 30 GPU hours, depending on the specific model architecture. For adapters and reverters, we start with a learning rate of $0.01$, reducing it by a factor of $0.1$ after the first epoch. These components are trained in pairs, requiring 1 to 5 GPU hours depending on the specific encoder and decoder architectures.

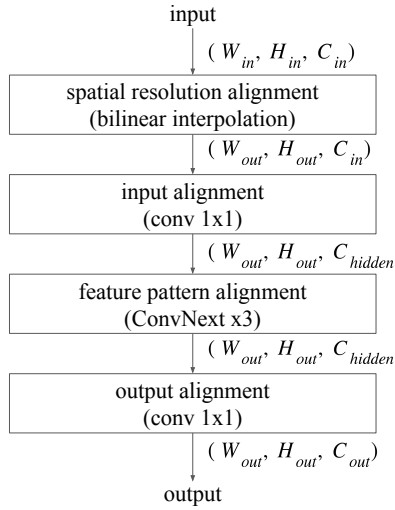

Figure A1: Architecture of adapter and reverter.

**Adapter and Reverter's Architecture.** We use the same architectures for both adapters and reverters across all CP models, as visualized in Figure A1. The dimension of the broadcasting feature map is set to $(128, 128, 64)$. $C_{\text{hidden}}$ is set to be 64. $W_{\text{in}}, H_{\text{in}}, C_{\text{in}}, W_{\text{out}}, H_{\text{out}},$ and $C_{\text{out}}$ of adapters and reverters vary according to the feature dimensions of each local model and the broadcasting feature map dimension. For instance, in the task- and model-agnostic setting, Agent 1's feature dimension is $128 \times 128 \times 64$, so we set $(W_{\text{in}}, H_{\text{in}}, C_{\text{in}}) = (W_{\text{out}}, H_{\text{out}}, C_{\text{out}}) = (128, 128, 64)$ for both its adapter and reverter. For Agent 2, with a feature dimension of $64 \times 64 \times 256$, we configure the adapter with $(W_{\text{in}}, H_{\text{in}}, C_{\text{in}}) = (64, 64, 256)$ and $(W_{\text{out}}, H_{\text{out}}, C_{\text{out}}) = (128, 128, 64)$, while the reverter is set with $(W_{\text{in}}, H_{\text{in}}, C_{\text{in}}) = (128, 128, 64)$ and $(W_{\text{out}}, H_{\text{out}}, C_{\text{out}}) = (64, 64, 256)$.

Table A1: Modality, encoder, and encoder parameters (M) of each heterogeneous model in the 3D object detection setting.

| Index | Agent 1 | Agent 2 | Agent 3 | Agent 4 |
|---|---|---|---|---|
| **Modality** | Lidar | Lidar | Camera | Camera |
| **Encoder** | PointPillar (Lang et al., 2019) | SECOND (Yan et al., 2018) | EfficientNetB0 (Tan, 2019) | ResNet101 (He et al., 2016) |
| **Encoder Param.(M)** | 0.87 | 3.79 | 56.85 | 6.88 |
| **Index** | **Agent 5** | **Agent 6** | **Agent 7** | **Agent 8** |
| **Modality** | Camera | Lidar | Camera | Lidar |
| **Encoder** | ResNet34 (He et al., 2016) | VoxelNet (Zhou & Tuzel, 2018) | EfficientNetB1 (Tan, 2019) | PointPillar (large) (Lang et al., 2019) |
| **Encoder Param.(M)** | 6.51 | 2.13 | 66.41 | 1.91 |
| **Index** | **Agent 9** | **Agent 10** | **Agent 11** | **Agent 12** |
| **Modality** | Camera | Lidar | Camera | Lidar |
| **Encoder** | ResNet50 (He et al., 2016) | SECOND (large) (Yan et al., 2018) | EfficientNetB2 (Tan, 2019) | VoxelNet (large) (Zhou & Tuzel, 2018) |
| **Encoder Param.(M)** | 6.88 | 4.82 | 71.43 | 3.18 |

**3D object detection setting.** Under the experiments on 3D object detection task, we prepared 12 heterogeneous models. Table A1 displays the Modality, Encoder, and Encoder Parameters (M) information of each of the 12 heterogeneous models. For model 7, 9, and 11, we enlarge the encoders by increasing the size of hidden layers. For all heterogeneous models, we choose pyramid fusion layers proposed by Lu et al. (2024) to be the fusion module and three $1 \times 1$ convolutional layers for classification, regression, and direction, respectively.

## A.2 Architectural Comparison between Existing Frameworks

Figure A2 illustrated various frameworks that address heterogeneous CP. Late fusion simply combines agent outputs through post-processing. Calibrator (Xu et al., 2023a) enhances this approach by using calibrators to address domain gaps between heterogeneous agent outputs. End-to-end training, while effective, lacks scalability due to its requirement of re-training all agents' models. It also compromises security and task flexibility by shared fusion models and decoders. HEAL (Lu et al., 2024) improves upon this by fixing decoders and fusion models, re-training only the encoders, reducing training resources but still facing scalability issues due to the computational cost of encoder retraining as well as the security issue due to the shared fusion models and decoders. Our proposed framework, STAMP, introduces a novel approach using lightweight adapter and reverter pairs to align feature maps for collaboration. The lightweight nature of these components ensures scalability, while the maintenance of local fusion and decoders ensures both security and task agnosticism. This design effectively addresses the limitations of previous methods.

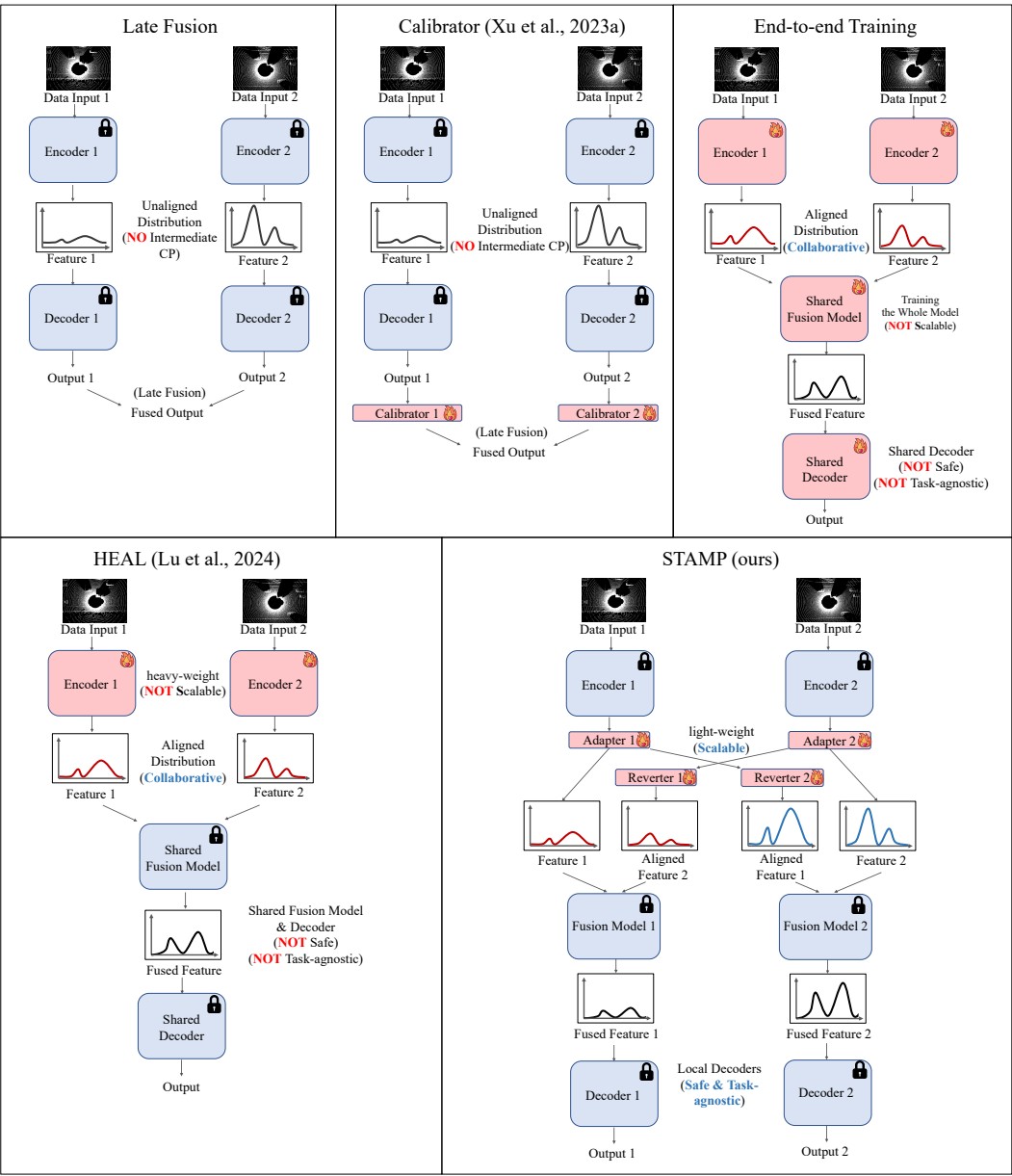

Figure A2: Architectural comparison of collaborative perception frameworks: existing approaches versus our proposed STAMP method. Blue boxes represent models with fixed parameters, while red boxes indicate models whose parameters are trained during the collaboration process.

## A.3 Additional Experiments

### A.3.1 Different Protocol Models

We conducted complementary experiments comparing different protocol model designs, analyzing variations in both encoder types and downstream tasks.

Table A2: Comparison of protocol designs, encoder types, and tasks across different agents.

| Protocol | Encoder Type | Protocol Task | Agent 1 (Lidar+Obj.) | Agent 2 (Cam.+Obj.) | Agent 3 (Lidar+Static Seg.) | Agent 4 (Lidar+Dyn. Seg.) |
|---|---|---|---|---|---|---|
| Non-Collab | - | - | 0.941 | 0.399 | 0.548 | 0.675 |
| STAMP | CNN-based | Object Det. | 0.936 ↓0.005 | 0.760 ↑0.362 | 0.624 ↑0.076 | 0.690 ↑0.014 |
| STAMP (ablations) | Camera-modality | Object Det. | 0.931 ↓0.010 | 0.777 ↑0.368 | 0.580 ↑0.032 | 0.671 ↓0.004 |
| | Camera + Lidar | Object Det. | 0.937 ↓0.004 | 0.762 ↑0.363 | 0.632 ↑0.084 | 0.714 ↑0.039 |
| | Point-transformer | Object Det. | 0.942 ↑0.001 | 0.775 ↑0.376 | 0.634 ↑0.086 | 0.696 ↑0.021 |
| | CNN-based | Dyn. Seg. | 0.935 ↓0.006 | 0.743 ↑0.344 | 0.624 ↑0.076 | 0.723 ↑0.048 |
| | CNN-based | Static. Seg. | 0.747 ↓0.194 | 0.412 ↑0.013 | 0.681 ↑0.133 | 0.235 ↓0.440 |

**Impact of Model Objectives** The experimental results shown in Table A2 demonstrate that the alignment's success significantly depends on the learning objectives between protocol models and agent architectures. When there is strong alignment between the protocol model and an agent's objectives, we observe performance improvements. For examples, A camera-modality protocol model improves camera-based Agent 2's performance from 0.760 to 0.777; a dynamic-segmentation protocol model enhances Agent 4's performance from 0.690 to 0.723. Similarly, a static-segmentation protocol model boosts Agent 3's performance from 0.624 to 0.681.

However, significant objective mismatches can lead to severe performance degradation. For instance, using a static-segmentation protocol model causes Agent 4's mAP to drop dramatically from 0.690 to 0.235. This highlights the importance of careful protocol model selection.

**Encoder Architecture Variations** While our baseline experiments primarily used CNN-based encoders, we explicitly tested different encoder architectures to understand their impact. As shown in our results table, we evaluated: CNN-based encoders and Point-transformer encoders.

The Point-transformer protocol model outperforms the original CNN-based protocol model, showing our framework's compatibility with different encoder architectures. Notably, the Point-transformer protocol model achieved slightly superior performance (AP@50 = 0.991) compared to its CNN-based counterpart (AP@50 = 0.973). This observation suggests an important insight: the overall performance of the protocol model is more crucial than its specific architectural design. In other words, a well-performing protocol model tends to benefit all agent types, regardless of their individual architectures.

### A.3.2 Adversarial Robustness Evaluation

Adversarial attacks involve intentionally perturbing inputs to mislead machine learning models while keeping the modifications nearly imperceptible(Tu et al., 2021; Gao et al., 2024b). Following James et al. Tu et al. (2021)'s collaborative white-box adversarial attack method with the same hyperparameters, we conducted adversarial attack experiments on the V2V4Real dataset with two agents per scene. We designated agent 1 as the attacker and agent 2 as the victim, comparing three settings:

- **End-to-end training:** Models trained end-to-end with full parameter access, enabling direct white-box attacks on the victim.
- **HEAL:** Agents share encoders but have different fusion models/decoders, assuming no victim model access.
- **STAMP:** Agents share no local models, using protocol representation for communication, assuming no victim model access.

The results shown in table A3 demonstrate that adversarial attacks have minimal impact on HEAL and STAMP frameworks due to local model security, while significantly degrading performance in

Table A3: Performance of different frameworks under adversarial attack on the V2V4Real dataset.

| AP@50 | End-to-end | HEAL | STAMP (ours) |
|---|---|---|---|
| Before Attack | 0.513 | 0.515 | 0.523 |
| After Attack | 0.087 | 0.506 | 0.503 |

end-to-end training where models are shared. This empirically supports our framework's robustness against malicious agent attacks.

### A.3.3 MORE COMPARISON WITH THE STATE-OF-THE-ART METHODS

We conducted additional experiments comparing with V2X-ViT(Xu et al., 2022a), CoBEVT(Xu et al., 2023c), HM-ViT(Xiang et al., 2023), HEAL(Lu et al., 2024) in a heterogeneous input modality setting. We configured four agents: two LiDAR agents using PointPillar and SECOND encoders, and two camera agents using EfficientNet-b0 and ResNet-101 encoders. For CoBEVT, HM-ViT, and HEAL, we followed their standard architecture and hyper-parameter setup. V2X-ViT does not support camera modality, so we follow HEAL to use ResNet-101 with Split-slat-shot for encoding RGB images to BEV features. For STAMP, we used pyramid fusion layers and three $1 \times 1$ convolutional layers (for classification, regression, and direction) across all heterogeneous models.

Table A4: Comparison of performance metrics (AP@50) for different models across heterogeneous agents.

| AP@50 | Agent 1 (PointPillar) | Agent 2 (SECOND) | Agent 3 (EfficientNet-B0) | Agent 4 ((ResNet-101)) | Average |
|---|---|---|---|---|---|
| V2X-ViT | - | - | - | - | 0.905 |
| CoBEVT | - | - | - | - | 0.899 |
| HM-ViT | - | - | - | - | 0.918 |
| HEAL | 0.971 | 0.958 | 0.776 | 0.771 | 0.934 |
| STAMP (ours) | 0.971 | 0.963 | 0.771 | 0.756 | 0.934 |

Note that CoBEVT, HM-ViT, and V2X-ViT use a single fusion layer and output layer for all modalities, while HEAL and our framework maintain separate fusion and output layers for each agent to preserve model independence. The reported accuracy is averaged across all samples.

### A.4 MULTI-GROUP AND MULTI-MODEL COLLABORATIONS SYSTEM

In our experimental findings, we observed a bottleneck effect in CP systems, where the overall system performance is constrained by the capabilities of the weakest agent. This limitation underscores the need for more selective collaboration, leading us to introduce the concept of a **Collaboration Group** - a set of agents that collaborate under specific criteria. These criteria are essential for maintaining the quality and integrity of CP, admitting agents that meet predefined standards while excluding those with inferior models, potential malicious intent, or incompatible alignments. As illustrated in Figure A3, we can distinguish between three collaborative system types:

- Single-group systems, where agents either operate independently or are compelled to collaborate with all others, are susceptible to performance bottlenecks caused by inferior agents and vulnerabilities introduced by malicious attackers.
- Multi-group single-model systems, allowing multiple collaboration groups but restricting agents to a single group because each agent can only equip a single model.
- Multi-group multi-model systems, enabling agents to join multiple groups if they meet the predefined standards.

The multi-group structure offers significant advantages over traditional single-group systems. It enhances agents' potential for diverse collaborations, consequently improving overall performance. This approach mitigates the bottleneck effect by allowing high-performing agents to maintain efficiency within groups of similar capability while potentially assisting less capable agents in other

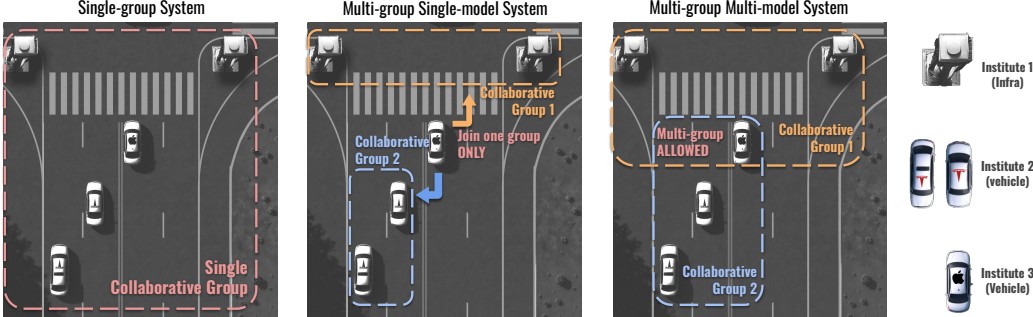

Figure A3: Comparison of collaborative perception systems: (Left) Single-group system where all agents collaborate within one group. (Middle) Multi-group single-model system allowing agents to join only one of multiple collaboration groups. (Right) Multi-group multi-model system enabling agents to participate in multiple collaboration groups simultaneously. The figure illustrates how different system architectures impact agent interactions and group formations in autonomous driving scenarios.

groups. Furthermore, it enhances system flexibility, enabling dynamic group formation based on specific task requirements or environmental conditions.

However, implementing such a multi-group system poses challenges for existing heterogeneous collaborative pipelines. End-to-end training approaches require simultaneous training of all models, conflicting with the concept of distinct collaboration groups. Methods like those proposed by Lu et al. (2024) require separate encoders for each group, becoming impractical as the number of groups increases due to computational and memory constraints.

Our proposed STAMP framework effectively addresses these limitations, offering a scalable solution for multi-group CP. The key innovation lies in its lightweight adapter and reverter pair (approximately 1MB) required for each collaboration group an agent joins. This efficient design enables agents to equip multiple adapter-reverter pairs, facilitating seamless participation in various groups without significant computational overhead. The minimal memory footprint ensures scalability, even as agents join numerous collaboration groups, making STAMP particularly well-suited for multi-group and multi-model collaboration systems.

## A.5 MORE VISUALIZATION RESULTS

Figure A4 and A5 illustrate more feature map and result visualizations before and after collaborative feature alignment (CFA). Prior to CFA, agents' feature maps exhibit disparate representations. For instance, in Figure A4, the pre-fusion feature maps of agents 1, 3, and 4 appear entirely black, indicating a significantly lower scale compared to agent 2's feature map. This discrepancy leads to instability in feature fusion. Post-CFA, the features are aligned to the same domain, resulting in more coherent fusion and accurate inference outputs.

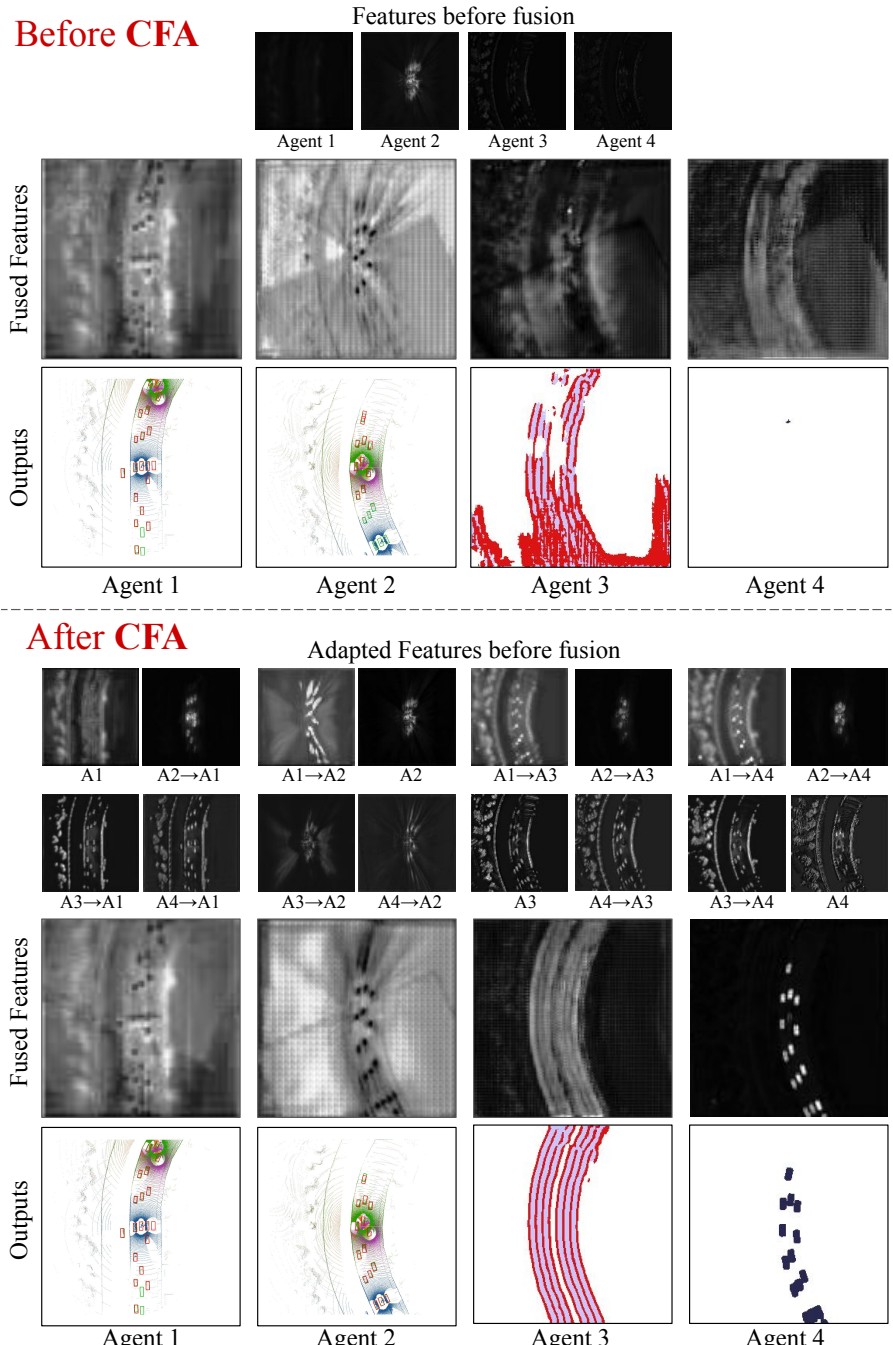

Figure A4: Visualization of feature maps and inference results before and after Collaborative Feature Alignment (CFA) in a three-agent scene. $A_i \rightarrow A_j$ denotes the feature map aligned from agent $i$'s domain to agent $j$'s domain, also represented as $F_{ij}$.

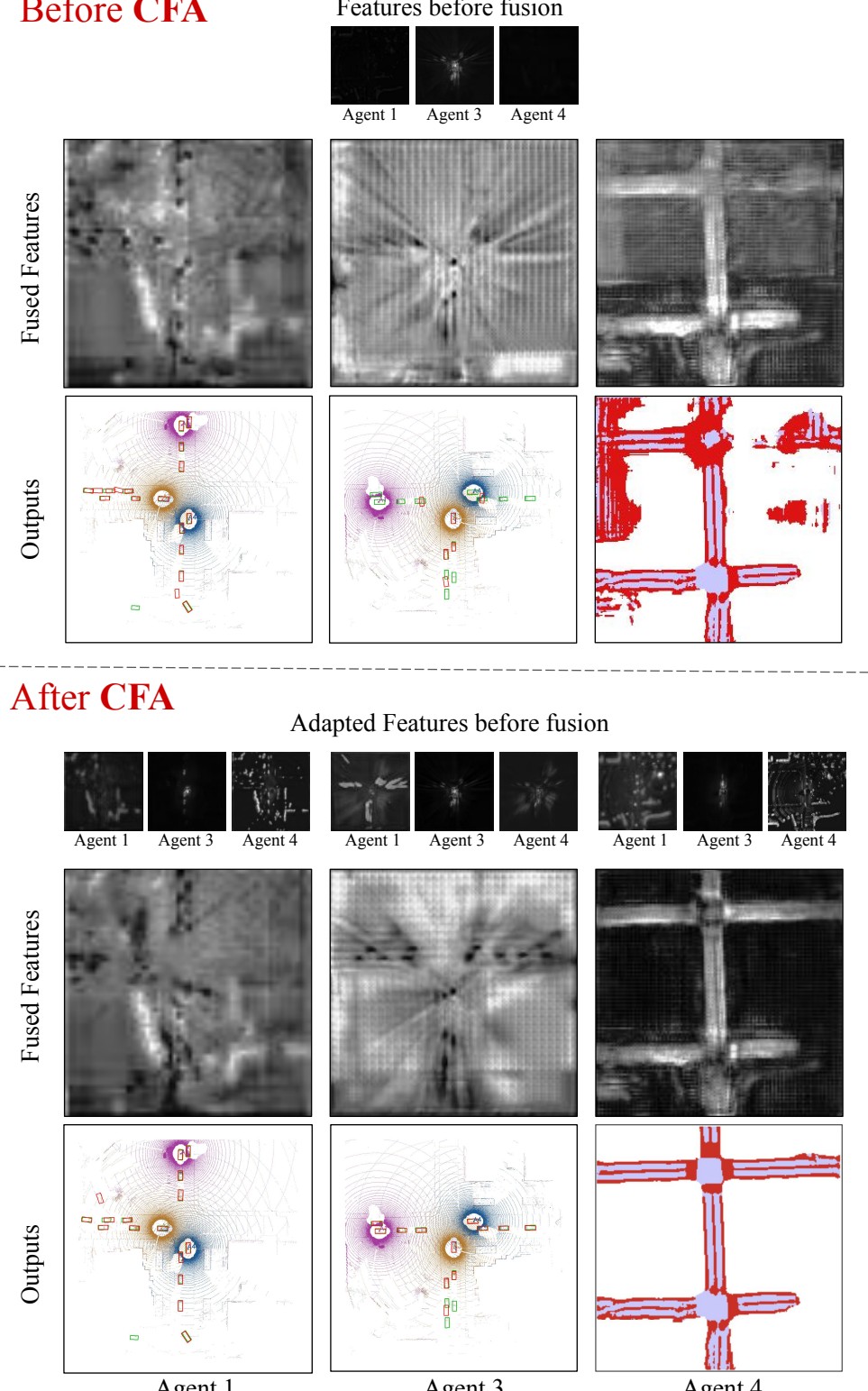

Figure A5: Visualization of feature maps and inference results before and after Collaborative Feature Alignment (CFA) in a four-agent scene. $A_i \rightarrow A_j$ denotes the feature map aligned from agent $i$'s domain to agent $j$'s domain, also represented as $F_{ij}$.

