# OpenReview forum: "STAMP: Scalable Task- And Model-agnostic Collaborative Perception"
_ICLR.cc/2025/Conference — ICLR 2025 Poster_

### Official Review · Reviewer_6FAr · 2024-10-22

**Soundness:** 3
**Presentation:** 3
**Contribution:** 2
**Rating:** 6
**Confidence:** 5

**Summary:**

This paper proposes STAMP, a task- and model-agnostic collaborative perception framework. The core idea is to first obtain a protocol BEV feature space, then align other local models' BEV features to this space using a simple DNN projection to achieve model agnosticism. A simple DNN is also used to map the aligned BEV features to a specific decoder and task head, achieving task agnosticism. Finally, experiments were conducted on OPV2V and V2V4Real.

**Strengths:**

1. This paper proposed a task- and modal-agnostic CP framework, which is new and the first work to address heterogeneous modalities, heterogeneous model architectures or parameters, and heterogeneous downstream tasks simultaneously.
2. The writing is good and fluent.

**Weaknesses:**

1. On line 227, the authors claim that the protocol model is not limited to any specific architecture or downstream task, making it a task- and model-agnostic framework. However, I disagree. The framework is task- and model-agnostic because it allows newly added agents to use different models or tasks, rather than due to the flexibility of the protocol model itself.
2. I think the author should compare more baseline methods, such as HM-ViT, DiscoNet, V2VNet, V2X-ViT, Where2comm, When2com, and What2com, not just compare with HEAL. I know your idea comes from HEAL, but comparing with other methods is necessary.
3. In Tab.2 and 3, I observe that STAMP achieves the best performance. However, I have some concerns about E2E training. The E2E training supposes to be the best, since it has the entire parameters to adapt the domain gap, which STAMP just use two projection DNN to adapt the features between different modalities and models, which is not make sense.
4. In Tab. 4, I find that the STAMP just has very little improvement even degradation. I don’t think there is much significance.
5. On line 52, the authors claim that this framework is robust against malicious agent attacks. However, they haven't proven this or conducted even a single experiment to support it. Moreover, I believe this claim is questionable. Although an attacker might not know the other agents' models, they could still inject malicious information into the protocol BEV features to attack the ego vehicle.

**Questions:**

1. In experiment, why use AP@30 and AP@50 rather than AP@50 and AP@70? I think AP@30 is not usually used in detection task.
2. Why not conduct experiments about the communication efficiency.
3. Since the task is different, how to align the decision space with different tasks (GT) in Sec. 3.3
4. For the feature space alignment, I don’t think it always works, some times it may has negative influence, because the BEV feature distribution is different across different agent. From Figure 5, we can see that the styles of different agents’ feature are not the same. As a result, just simply forcing the features to be same is not a good idea.
5. The current trend in autonomous driving models is towards increasing model size and vehicle computational power. Additionally, there is a shift towards end-to-end models, significantly enhancing the autonomous capabilities of individual vehicles. Given these advancements, how much market potential remains for multi-vehicle cooperative perception based on intermediate BEV feature communication? How does the author view this issue?

---

> ### Author Response · Authors · 2024-11-23
>
> Thank you for your detailed review and insightful comments. Please kindly see below for our responses to your comments:
>
> > ### In experiment, why use AP@30 and AP@50 rather than AP@50 and AP@70? I think AP@30 is not usually used in detection task.
>
> Here we provide the object detection experimental results for the AP@70 evaluation metric
>
> ### AP 70
>
> | $\sigma$         | Method           | Agent 1 | Agent 2 | Agent 3 | Agent 4 |
> |-------------------|------------------|---------|---------|---------|---------|
> | $\sigma=0.0$     | Late Fusion      | 0.846   | 0.862   | 0.869   | 0.871   |
> |                  | Calibrator       | 0.844   | 0.860   | 0.871   | 0.876   |
> |                  | E2E Training     | 0.826   | 0.951   | 0.947   | **0.966**   |
> |                  | HEAL             | 0.840   | 0.951   | **0.961**   | 0.964   |
> |                  | Ours             | **0.846** | **0.954** | **0.961** | 0.961 |
> | $\sigma=0.2$     | Late Fusion      | 0.842   | 0.852   | 0.865   | 0.868   |
> |                  | Calibrator       | 0.838   | 0.846   | 0.857   | 0.871   |
> |                  | E2E Training     | 0.825   | 0.921   | 0.934   | 0.952   |
> |                  | HEAL             | 0.838   | 0.938   | **0.948**   | **0.959**   |
> |                  | Ours             | **0.845** | **0.942** | 0.942 | 0.956 |
> | $\sigma=0.4$     | Late Fusion      | 0.799   | 0.820   | 0.821   | 0.825   |
> |                  | Calibrator       | 0.797   | 0.814   | 0.821   | 0.822   |
> |                  | E2E Training     | 0.808   | **0.902**   | 0.904   | 0.911   |
> |                  | HEAL             | 0.823   | 0.899   | 0.900   | 0.911   |
> |                  | Ours             | **0.838** | 0.893 | **0.906** | **0.921** |
>
> > ### Why not conduct experiments about the communication efficiency?
>
> Our framework employs an adapter mechanism to align local features with the protocol domain for inter-agent communication. This design offers inherent flexibility in terms of communication bandwidth, as it is not constrained to specific feature resolutions or channel sizes. While our main experiments utilize a consistent configuration (128×128 feature resolution with 64 channels), we conducted additional ablation studies with varying channel sizes to evaluate the framework's performance across different communication bandwidth settings. These experiments demonstrate our framework's adaptability to diverse bandwidth requirements, addressing potential concerns about various communcation bandwidth limits.
>
> There are some existing techniques for improving communication efficiency in multi-agent systems, including selective communication [3], tensor sparsification [4], and tensor codebook-based methods [1,2]. While these approaches have proven effective in homogeneous settings, their adaptation to heterogeneous multi-agent systems presents an interesting opportunity. Specifically, integrating these communication-efficient techniques into our framework could potentially yield significant improvements in bandwidth utilization on heterogenous collaboration. This intersection of communication efficiency and heterogeneous multi-agent systems represents a promising direction for future research.
>
> [1] Hu, Y., Peng, J., Liu, S., Ge, J., Liu, S., & Chen, S. (2024). Communication-Efficient Collaborative Perception via Information Filling with Codebook. In Proceedings of the IEEE/CVF Conference on Computer Vision and Pattern Recognition (pp. 15481-15490).
>
> [2] Hu, Y., Pang, X., Qin, X., Eldar, Y. C., Chen, S., Zhang, P., & Zhang, W. (2024). Pragmatic Communication in Multi-Agent Collaborative Perception. arXiv preprint arXiv:2401.12694.
>
> [3] Liu, Y. C., Tian, J., Ma, C. Y., Glaser, N., Kuo, C. W., & Kira, Z. (2020, May). Who2com: Collaborative perception via learnable handshake communication. In 2020 IEEE International Conference on Robotics and Automation (ICRA) (pp. 6876-6883). IEEE.
>
> [4] Hu, Y., Fang, S., Lei, Z., Zhong, Y., & Chen, S. (2022). Where2comm: Communication-efficient collaborative perception via spatial confidence maps. Advances in neural information processing systems, 35, 4874-4886.

---

> > ### Author Response · Authors · 2024-11-23
> >
> > > ### Since the task is different, how to align the decision space with different tasks (GT) in Sec. 3.3.
> >
> > The alignment of decision spaces across different tasks is achieved based on Equation (11). Let us break down the alignment process:
> >
> > **Protocol Domain Alignment**
> >
> > We first align $F_{iP}^{1:K}$ with $F_{P}^{1:K}$, where $F^{1:K} = (F^1, \cdots, F^K)$ represents features across $K$  world states $\mathbf{x}^{1:K} = (\mathbf{x}^1, \cdots, \mathbf{x}^K)$. Since both feature sets exist in the protocol domain and are derived from the same world states $\mathbf{x}^{1:K}$, we expect them to produce the same outputs when processed through the protocol model. This alignment is enforced using the protocol model's loss function $\mathcal{L}_P$ against its ground truth $\text{GT}_P$. For instance, with a BEV segmentation protocol model, $\text{GT}_P$ would represent segmentation labels, and $\mathcal{L}_P$ would be an appropriate segmentation loss function.
> >
> > **Agent-Specific Domain Alignment**
> >
> > Similarly, we align both $F_{ii}^{1:K}$ with $F_i^{1:K}$ and $F_{Pi}^{1:K}$ with $F_i^{1:K}$. These feature sets exist in agent $i$'s domain and are derived from the same world state $\mathbf{x}^{1:K}$. The alignment is achieved by passing these features through model $i$ and comparing the outputs against model $i$'s ground truth.
> >
> > For the complete mathematical formulation of these alignments, please refer to Equation (11) in the main paper.
> > Due to the equation length limit of this page, we cannot write long equation here. For the complete mathematical formulation of these alignments, please refer to Equation (11) in the main paper.

---

> > > ### Author Response · Authors · 2024-11-23
> > >
> > > > ### For the feature space alignment, I don’t think it always works, some times it may has negative influence, because the BEV feature distribution is different across different agent. From Figure 5, we can see that the styles of different agents’ feature are not the same. As a result, just simply forcing the features to be same is not a good idea.
> > >
> > > We understand the concern of the capability of the feature space alignment. The heterogeneity of agents—arising from differences in input modalities, model architectures, and downstream tasks—naturally leads to variations in BEV feature distributions (styles). The whole collaborative feature alignment (CFA) process, including **feature space alignment** and **decision space alignment**, is designed to addresses this challenge.
> > >
> > > First, we observed that feature space alignment alone is insufficient to bridge these distributional differences. This led us to introduce the decision space alignment loss as a complementary mechanism. While we initially hypothesized that feature space alignment might have negative impacts after seeing improvements from decision space alignment, our ablation studies revealed otherwise. As demonstrated in Figures 3(C) and 4(C), **removing the feature space alignment loss significantly degrades performance, leading us to retain both alignment losses in our final framework.**
> > >
> > > We displayed some more feature map visualization results on the Appendix section. Taking Figure A4 as an example, before CFA, the features before fusion are dramatically different in styles. (The feature maps of Agent 1, 3, 4 look purely black because the value are too small comparing the feature map of Agent 2.) After CFA, the feature are much more aligned. It is non-negotiable that the features from different heterogeneous agents are not perfectly aligned visually, but experimental results on Table 2 and Table 4 reveals that the decision space alignment loss enables high-quality outputs despite the feature space differences.
> > >
> > > We believe further improving our current collaborative feature alignment method is a important for the future research.

---

> > > > ### Author Response · Authors · 2024-11-23
> > > >
> > > > > ### The current trend in autonomous driving models is towards increasing model size and vehicle computational power. Additionally, there is a shift towards end-to-end models, significantly enhancing the autonomous capabilities of individual vehicles. Given these advancements, how much market potential remains for multi-vehicle cooperative perception based on intermediate BEV feature communication? How does the author view this issue?
> > > >
> > > > We appreciate this thoughtful question about the future of multi-vehicle cooperative perception. Let us share our perspectives.
> > > >
> > > > First, it's important that multi-vehicle collaboration and the trend toward more powerful individual vehicles are not competing approaches. Instead of choosing one over the other, we believe these technologies should develop hand-in-hand. While larger models and end-to-end systems make individual vehicles smarter, cooperative perception helps them work together more effectively.
> > > >
> > > > Second, consider the different problems these approaches solve. More powerful individual vehicle systems help AV reach or exceed human cognitive abilities—like understanding complex traffic scenarios or making decisions. However, multi-vehicle collaboration helps vehicles overcome physical limitations. Take the example of an occluded pedestrian about to cross the street. No matter how advanced a single vehicle's AI system is, it simply cannot "see" through other vehicles or buildings. This is where V2X systems shine, as they allow vehicles to share what they see with others, creating a much safer driving environment.
> > > >
> > > > Looking to the future, we think that V2X research, especially systems using BEV feature communication, is still in its early phases. Multi-vehicle cooperative perception based on intermediate BEV feature communication is one promising direction and it will be intergrated to end-to-end system. There are many unsolved problem such as communication efficiency, communication latency, adversarial robustness, agent heterogeneity, etc. While we don't expect to see this methods to be deployed on roads in large scale within one or two years, we strongly believe this research direction will play a crucial role in the future of autonomous driving.

---

> ### Author Response · Authors · 2024-11-24
>
> > ### Weakness 3: In Tab.2 and 3, I observe that STAMP achieves the best performance. However, I have some concerns about E2E training. The E2E training supposes to be the best, since it has the entire parameters to adapt the domain gap, which STAMP just use two projection DNN to adapt the features between different modalities and models, which is not make sense.
>
> This is a very good point. We also have this confusion for the first time and tried to investigate some possible reasons that cause E2E training under-perform STAMP.
>
> We hypothesize that our method's superiority may stem from its ability to accommodate varying convergence rates among different models. By training models separately, we can select optimal checkpoints for each model based on individual validation performance. In contrast, end-to-end training necessitates choosing a single checkpoint that may not be optimal for all models simultaneously. We inspected the validation losses during training, we observed tendencies of overfitting in LiDAR models and under-fitting in camera models in end-to-end training.
>
> The validation loss in training time is listed as follow. Notice that we trained the end-to-end model for 120 epochs, four times as training each agent, because the number of parameters of the end-to-end model is roughly equal to the summation of all four models.
>
> | Epoch   | 4    | 8    | 12   | 16   | 20   | ...  | 80     | 112  | 116  | 76   | 80   | 84   | 88   | 92   | 96   | 100  | 104  | 108  | 112  | 116  |
> |-----------|------|------|------|------|------|------|------|------|------|------|------|------|------|------|------|------|------|------|------|------|
> | Agent 1 | 0.59 | 0.43 | **0.19** | 0.28 | 0.24 | ...  |  -    | -    | -    | -    | -    | -    | -    | -    | -    | -    | -    | -    | -    | -    |
> | Agent 2 | 0.56 | 0.48 | **0.23** | 0.29 | 0.25 | ...  |  -    | -    | -    | -    | -    | -    | -    | -    | -    | -    | -    | -    | -    | -    |
> | Agent 3 | 0.72 | 0.71 | 0.54 | 0.50 | **0.44** | ...  |  -    | -    | -    | -    | -    | -    | -    | -    | -    | -    | -    | -    | -    | -    |
> | Agent 4 | 0.74 | 0.65 | 0.54 | 0.48 | **0.43** | ...  |  -    | -    | -    | -    | -    | -    | -    | -    | -    | -    | -    | -    | -    | -    |
> | End2End  | 0.85 | 0.78 | 0.85 | 0.72 | 0.67 | ...   | 0.31 | 0.28 | 0.26 | 0.27 | 0.36 | 0.32 | 0.25 | 0.29 | **0.24** | 0.34 | 0.27 | 0.31 | 0.28 | 0.26 | 0.27 | 0.36 | 0.32 |
>
> While these observations are intriguing, comprehensive experiments to validate these conjectures were beyond the scope of this work. Future research should focus on developing a theoretical analysis to explain these phenomena and their impact on collaborative perception performance.

---

> ### Comment · Reviewer_6FAr · 2024-11-24
>
> Dear authors,
>
> Aside from the questions I raised, you should also reply to the points of weakness.
>
> Thanks,
> Reviewer 6FAr

---

> > ### Author Response · Authors · 2024-11-25
> >
> > Thank you for your follow-up note. We are currently conducting additional experiments to thoroughly address each weakness point raised in your review. We will provide comprehensive responses to all points of weakness along with supporting experimental results in our revision. We appreciate your patience and valuable feedback.
> >
> > > ### Weakness 1: On line 227, the authors claim that the protocol model is not limited to any specific architecture or downstream task, making it a task- and model-agnostic framework. However, I disagree. The framework is task- and model-agnostic because it allows newly added agents to use different models or tasks, rather than due to the flexibility of the protocol model itself.
> >
> > We appreciate the reviewer's observation and agree with this point. Indeed, we need to clarify that it is not the protocol model itself, but rather the alignment process that makes our framework task- and model-agnostic. As stated in our introduction, "the alignment process is designed to be task- and model-agnostic, allowing our framework to integrate with various models and tasks without retraining the model or the need to share models among agents." This is further reinforced in Section 3.2, where we note that "Our proposed framework, STAMP, enables collaboration among existing heterogeneous agents without sharing model details or downstream task information." We thank the reviewer for bringing this distinction to our attention, and we will carefully review the manuscript to ensure consistent and precise language throughout.

---

> ### Author Response · Authors · 2024-11-25
>
> > ### Weakness 5: On line 52, the authors claim that this framework is robust against malicious agent attacks. However, they haven't proven this or conducted even a single experiment to support it. Moreover, I believe this claim is questionable. Although an attacker might not know the other agents' models, they could still inject malicious information into the protocol BEV features to attack the ego vehicle.
>
> Thank you for raising this important point about security analysis. We would like to clarify the possibility of white-box adversarial attacks in our framework.
>
> The traditional white-box attack assumption requires full access to model parameters to propagate gradients from the supervision to the target tensor. However, in STAMP, while agents have access to the protocol model, they do not have access to other agents' fusion and output layers, so the gradient of victim models cannot be accessed. Let us illustrate this through STAMP's pipeline:
>
> 1. Encoding: $F_j = E_j(I_j)$
>
> 2. Adaptation:  $F_{jP} = \upphi_j(F_j), \quad \forall i \in \\{1, 2, \ldots, N\\}$
>
> 3. \begin{equation}
> \text{Reversion:}\ F_{ji} =
> \begin{cases}
> \uppsi_i(F_{jP}), \text{if } j \neq i,\quad \\
> F_j, \text{if } j = i
> \end{cases}
> \quad \forall j, i \in \\{1, 2, \ldots, N\\}
> \end{equation}
>
> 4. Fusion: $F_i' = U_i(\\{ F_{ji} \mid \mathcal{N}(i, j) \leq \delta \\})$
>
> 5. Decoding: $O_i = D_i(F'_i)$
>
> Consider a scenario where model $i$ attempts to attack model $j$. In a white-box attack setting, we would supervise output $O_i$ and aim to propagate gradients to $F_j$. The gradient computation involves layers $D_i$, $U_i$, $\uppsi_i$, and $\upphi_j$. Since only $\upphi_j$ belongs to model $j$ while all other layers belong to model $i$, it failing to meet the requirements for an ideal white-box attack.
>
> To empirically validate this analysis, we conducted adversarial attack experiments on the object detection task (following the setup in section 4.2). We chose object detection due to time constraints during rebuttal and HEAL's limitation to homogeneous tasks. Following James et al. [1]'s collaborative white-box adversarial attack method with identical hyperparameters, we tested on the V2V4Real dataset with two agents per scene. We designated agent 1 as the attacker and agent 2 as the victim, comparing three settings:
>
> 1. End-to-end training: Models trained end-to-end with full parameter access, enabling direct white-box attacks on the victim.
>
> 2. HEAL: Agents share encoders but have different fusion models/decoders, assuming no victim model access.
>
> 3. STAMP: Agents share no local models, using protocol representation for communication, assuming no victim model access.
>
> Results:
>
> | AP@50          | End-to-end | HEAL  | STAMP (ours) |
> |-----------------|-------------|-------|--------------|
> | Before Attack   | 0.513       | 0.515 | 0.523        |
> | After Attack    | 0.087       | 0.506 | 0.503        |
>
> The results demonstrate that adversarial attacks have minimal impact on HEAL and STAMP frameworks due to local model security, while significantly degrading performance in end-to-end training where models are shared. This empirically supports our framework's robustness against malicious agent attacks.
>
> The results demonstrate that adversarial attacks have minimal impact on HEAL and STAMP frameworks due to local model security, while significantly degrading performance in end-to-end training where models are shared. We understand that security is a large topic that requires extensive experiments and analysis. Due to time constraints, we only conducted these initial experiments. We believe comprehensively evaluate and analyze the adversarial robustness in heterogeneous collaborative perception is important for the future research.
>
> [1] Tu, J., Wang, T., Wang, J., Manivasagam, S., Ren, M., & Urtasun, R. (2021). Adversarial attacks on multi-agent communication. In Proceedings of the IEEE/CVF International Conference on Computer Vision (pp. 7768-7777).

---

> ### Author Response · Authors · 2024-11-25
>
> > ### Weakness 2: I think the author should compare more baseline methods, such as HM-ViT, DiscoNet, V2VNet, V2X-ViT, Where2comm, When2com, and What2com, not just compare with HEAL. I know your idea comes from HEAL, but comparing with other methods is necessary.
>
> Thank you for this valuable suggestion regarding baseline comparisons. We would like to explain our baseline selection rationale:
>
> - Our work focuses on collaborative perception with heterogeneous models. Methods such as DiscoNet[1], V2VNet[2], V2X-ViT[3], Where2comm[4], When2com[5], and What2comm[6] are designed for homogeneous collaboration and cannot support heterogeneous models, making direct comparisons challenging.
>
> - While CoBEVT[7] and HM-ViT[8] supports heterogeneous input modalities, our framework addresses a different aspect of heterogeneity - it enables collaboration among existing heterogeneous models without requiring model redesign or retraining.
>
> - HEAL[9] represents the current state-of-the-art in heterogeneous collaborative perception, making it the most relevant and representative baseline for evaluating our framework's effectiveness.
>
> Nevertheless, following your suggestion, we conducted additional experiments comparing with V2X-ViT, CoBEVT, HM-ViT, HEAL in a heterogeneous input modality setting. We configured four agents: two LiDAR agents using PointPillar and SECOND encoders, and two camera agents using EfficientNet-b0 and ResNet-101 encoders. For CoBEVT, HM-ViT, and HEAL, we followed their standard architecture and hyper-parameter setup. V2X-ViT does not support camera modality, so we follow HEAL to use ResNet-101 with Split-slat-shot for encoding RGB images to BEV features. For STAMP, we used pyramid fusion layers and three 1×1 convolutional layers (for classification, regression, and direction) across all heterogeneous models.
>
> Results on the OPV2V dataset:
> | AP@50           | Agent 1 (PointPillar) | Agent 2 (SECOND) | Agent 3 (EfficientNet-B0) | Agent 4 (ResNet-101) | Average |
> |------------------|-----------------------|------------------|--------------------------|-----------------------|---------|
> | V2X-ViT [3] | - | - | - | - | 0.905 |
> | CoBEVT [7]          | -                     | -                | -                        | -                     | 0.899   |
> | HMViT [8]          | -                     | -                | -                        | -                     | 0.918   |
> | HEAL [10]           | 0.971                 | 0.958            | 0.776                    | 0.771                 | 0.934   |
> | STAMP (ours)    | 0.971                 | 0.963            | 0.771                    | 0.756                 | 0.934   |
>
> Note: CoBEVT, HM-ViT, and V2X-ViT uses a single fusion layer and output layer for all modalities, while HEAL and our framework maintains separate fusion and output layers for each agent to preserve model independence. The reported accuracy is averaged across all samples.
>
> References:
>
> [1] Li et al. (2021). Learning distilled collaboration graph for multi-agent perception. NeurIPS, 34, 29541-29552.
>
> [2] Wang et al. (2020). V2vnet: Vehicle-to-vehicle communication for joint perception and prediction. ECCV, 605-621.
>
> [3] Xu et al. (2022). V2x-vit: Vehicle-to-everything cooperative perception with vision transformer. ECCV, 107-124.
>
> [4] Hu et al. (2022). Where2comm: Communication-efficient collaborative perception via spatial confidence maps. NeurIPS, 35, 4874-4886.
>
> [5] Liu et al. (2020). When2com: Multi-agent perception via communication graph grouping. CVPR, 4106-4115.
>
> [6] Yang et al. (2023). What2comm: Towards communication-efficient collaborative perception via feature decoupling. ACM MM, 7686-7695.
>
> [7] Xu et al. (2023). CoBEVT: Cooperative Bird’s Eye View Semantic Segmentation with Sparse Transformers. CoRL, 989-1000.
>
> [8] Xiang et al. (2023). HM-ViT: Hetero-modal vehicle-to-vehicle cooperative perception with vision transformer. ICCV, 284-295.
>
> [9] Lu et al. (2024). An extensible framework for open heterogeneous collaborative perception. ICLR.

---

> ### Author Response · Authors · 2024-11-25
>
> > ### Weakness 4: In Tab. 4, I find that the STAMP just has very little improvement even degradation. I don’t think there is much significance.
>
> Thank you for this observation about the performance improvements. While some improvements may appear modest, our results demonstrate several key achievements:
>
> As detailed in the paper, our method consistently outperforms single-agent performance for agents 3 and 4 in the BEV segmentation task, and achieves substantial gains for agent 2's camera-based 3D object detection (improving AP@50 from 0.399 to 0.760 in noiseless conditions). Importantly, when compared to collaboration without feature alignment, which significantly degrades performance below single-agent baselines, our approach maintains or improves performance across most agents.
>
> We acknowledge the performance decrease observed with Agent 1 compared to its single-agent baseline. As explained in the paper, this is attributed to a fundamental challenge in collaborative systems - the "bottleneck effect" where a weaker agent (in this case, Agent 2 with less accurate camera sensors for 3D object detection) can constrain the overall system performance.
>
> This observation has led us to introduce the concept of a Multi-group Collaboration System in the appendix section, which we believe will effectively address these performance variations. As the first framework enabling task- and model-agnostic collaborative perception, STAMP establishes a foundation for heterogeneous collaboration while maintaining local model independence and security. Moving forward, we identify several important research directions: optimizing the STAMP framework and multi-group collaboration system, improving collaboration efficiency, alleviating the "bottleneck effect", and further enhancing and evaluating system security. These aspects represent crucial areas for future investigation in heterogeneous collaborative perception.

---

> ### Author Response · Authors · 2024-11-27
> **Response to Reviewer 6FAr**
>
> Thanks for taking the time to provide your valuable feedback. We have carefully addressed all of your concerns and believe that our responses have fully resolved the issues you raised. With the discussion period ending soon, we kindly request that you review our responses at your convenience. Please let us know if you have any further questions or require additional clarification—we are more than willing to provide any additional information needed. Thanks again for your time and consideration.

---

> > ### Comment · Reviewer_6FAr · 2024-12-02
> >
> > The author addressed most of my concerns, I will raise my score.

---

> ### Author Response · Authors · 2024-12-02
> **Understanding Why End-to-End Training Underperforms STAMP**
>
> We recently discovered that the HEAL framework reached similar conclusions regarding end-to-end training. In their open review discussion[1], they observed that collaborative training (which we refer to as end-to-end training) can result in unbalanced and insufficient training when handling multiple agent types. These observations align with the patterns demonstrated in our training logs[2].
>
> [1] https://openreview.net/forum?id=KkrDUGIASk&noteId=WsaNjkXldg
>
> [2] https://openreview.net/forum?id=8NdNniulYE&noteId=XtGnXXjHWi

---

> ### Author Response · Authors · 2024-12-03
>
> We sincerely appreciate the thoughtful reviews and helpful suggestions that have strengthened our work.

---

### Official Review · Reviewer_m4Vi · 2024-11-01

**Soundness:** 2
**Presentation:** 3
**Contribution:** 3
**Rating:** 6
**Confidence:** 4

**Summary:**

The manuscript proposed a collaborative perception framework for heterogeneous agents, highlighting its feature of task- and model-agnostic. The framework contains a lightweight adapter-reverter pair, transforming the features between agent-specific domains and a shared protocol domain. The framework is tested on both simulated(OPV2V) and real-world (V2V4Real) datasets.

**Strengths:**

1. The idea of using an adapter-reverter pair in collaborative perception system for heterogeneous agents is intuitive. The design of the adapter & reverter is light-weight and does support solving most of the issue caused by heterogeneous encoders, such as resolution feature dimension.

2. The experimental results validate the effectiveness of the framework and prove it is task- and model-agnostic. More surprisingly, the training efficiency is significantly higher than the existing methods.

3. The manuscript organizes very well and the visualization is clear and easy to understand.

**Weaknesses:**

In the methodology section, the authors propose using the L2 norm as the training loss to align the agent-specific features with the protocol features. However, to fully understand this approach, more information on the architecture and capability of the protocol model is needed. Knowledge-distillation designs like this can sometimes risk alignment failure if there is a significant capability gap between the models. This may also explain the limitation noted in A.3, where the system performance is constrained by the weakest agent.

This reviewer is concerned that this may pose a drawback, as finding a suitable protocol model that meets the requirements of various modern encoders could be challenging. To address this concern, it would be helpful if the authors can provide more details about criteria of the protocol model selection, the current protocol model architecture, model size and its capabilities like task performances, as well as the comparison with those of the agent models.

**Questions:**

1. Following the weakness section, have authors considered the risk of alignment failure when there are significant capability differences between models? How do you pick the protocol model to maintain the robustness of the framework?

2. Another follow-up to the weaknesses section, this reviewer noticed that, in the experimental setup, all encoders are CNN-based. Has the author tried different combinations of protocol models and agent encoders, such as using a transformer-based protocol model while allowing agents to have either CNN-based or transformer-based encoders or other combinations.

---

> ### Author Response · Authors · 2024-11-27
>
> > ### Question 1: Following the weakness section, have authors considered the risk of alignment failure when there are significant capability differences between models? How do you pick the protocol model to maintain the robustness of the framework?
>
> > ### Question 2: Another follow-up to the weaknesses section, this reviewer noticed that, in the experimental setup, all encoders are CNN-based. Has the author tried different combinations of protocol models and agent encoders, such as using a transformer-based protocol model while allowing agents to have either CNN-based or transformer-based encoders or other combinations.
>
> We thank the reviewer for these insightful questions about model capability differences and encoder architectures. To address these concerns, we conducted complementary experiments comparing different protocol model designs, analyzing variations in both encoder types and downstream tasks.
>
> | Protocol                | Encoder Type           | Protocol Task      | Agent 1 (lidar+obj.)          | Agent 2 (cam.+obj.)        | Agent 3   (lidar+static. seg.)          | Agent 4 ( lidar+dyn. seg.)         |
> |-------------------------|------------------------|--------------------|-------------------|-------------------|-------------------|-------------------|
> | Non-Collab             | -                      | -                  | 0.941            | 0.399            | 0.548            | 0.675            |
> | STAMP                  | CNN-based              | Object Det.        | 0.936 (−0.005)   | 0.760 (+0.362)   | 0.624 (+0.076)   | 0.690 (+0.014)   |
> | STAMP (ablations)      | Camera-modality        | Object Det.        | 0.931 (−0.010)   | 0.777 (+0.368)   | 0.580 (+0.032)   | 0.671 (-0.004)   |
> | | Camera + Lidar | Object Det.       | 0.937 (-0.004)   | 0.762 (+0.363)   | 0.632 (+0.084)   | 0.714 (+0.039)   |
> |                        | Point-transformer      | Object Det.        | 0.942 (+0.001)   | 0.775 (+0.376)   | 0.634 (+0.086)   | 0.696 (+0.021)   |
> |                        | CNN-based              | Dynamic Obj. Seg.  | 0.935 (−0.006)   | 0.743 (+0.344)   | 0.624 (+0.076)   | 0.723 (+0.048)   |
> |                        | CNN-based              | Static Obj. Seg.   | 0.747 (-0.194)   | 0.412 (+0.013)   | 0.681 (+0.133)   | 0.235 (-0.440)   |
>
> **Impact of Model Capability Differences**
>
> Our experiments demonstrate that the alignment's success significantly depends on the compatibility between protocol models and agent architectures. When there is strong alignment between the protocol model and an agent's capabilities, we observe performance improvements:
>
> - A camera-modality protocol model improves camera-based Agent 2's performance from 0.760 to 0.777
> - A dynamic-segmentation protocol model enhances Agent 4's performance from 0.690 to 0.723
> - A static-segmentation protocol model boosts Agent 3's performance from 0.624 to 0.681
>
> However, significant capability mismatches can lead to severe performance degradation. For instance, using a static-segmentation protocol model causes Agent 4's mAP to drop dramatically from 0.690 to 0.235. This highlights the importance of careful protocol model selection.
>
> Based on these findings, we believe that using a "task-agnostic model," such as the scene completion model proposed by Li et al. [1], could help mitigate these alignment challenges and enhance framework robustness. This approach represents a promising direction for future research to address the capability differences.
>
> **Encoder Architecture Variations**
>
> While our baseline experiments primarily used CNN-based encoders, we explicitly tested different encoder architectures to understand their impact. As shown in our results table, we evaluated: CNN-based encoders and Point-transformer encoders.
>
> The Point-transformer protocol model outperforms the original CNN-based protocol model, showing our framework's compatibility with different encoder architectures. Notably, the Point-transformer protocol model achieved slightly superior performance (AP@50=0.991) compared to its CNN-based counterpart (AP@50=0.973). This observation suggests an important insight: the overall performance of the protocol model is more crucial than its specific architectural design. In other words, a well-performing protocol model tends to benefit all agent types, regardless of their individual architectures.
>
> However, while our initial results are promising, we acknowledge that a more comprehensive analysis of architectural choices and their impacts would be valuable for the research community. This includes investigating a broader range of encoder architectures and understanding the nuances of how protocol model performance translates to agent collaboration effectiveness. We consider this an important direction for future research.
>
> [1] Li et al. (2023). Multi-robot scene completion: Towards task-agnostic collaborative perception. CoRL, 2062–2072.

---

> ### Author Response · Authors · 2024-11-27
>
> > ### Weakness 1: In the methodology section, the authors propose using the L2 norm as the training loss to align the agent-specific features with the protocol features. However, to fully understand this approach, more information on the architecture and capability of the protocol model is needed. Knowledge-distillation designs like this can sometimes risk alignment failure if there is a significant capability gap between the models. This may also explain the limitation noted in A.3, where the system performance is constrained by the weakest agent.
>
> > ### Weakness 2: This reviewer is concerned that this may pose a drawback, as finding a suitable protocol model that meets the requirements of various modern encoders could be challenging. To address this concern, it would be helpful if the authors can provide more details about criteria of the protocol model selection, the current protocol model architecture, model size and its capabilities like task performances, as well as the comparison with those of the agent models.
>
> We appreciate the reviewer's concerns regarding protocol model selection and potential capability gaps. Let us clarify our protocol model architecture and address the alignment considerations.
>
> **Protocol Model Architecture**
>
> For our main experiments, we utilized an architecture identical to Agent 2 from the heterogeneous collaborative 3D object detection experiments in Section 4.2. To maintain heterogeneity in model parameters while preserving architectural consistency, we initialized the protocol model with different random seeds. We acknowledge this should have been explicitly stated in the paper and will include these details in the final version.
>
> **Protocol Model Selection and Performance**
>
> Our supplementary experiments with different protocol model architectures revealed important insights about the STAMP framework:
>
> - The framework demonstrates resilience to variations in protocol model architecture, suggesting flexibility in architectural design choices.
> - Performance correlates more strongly with training objectives (downstream tasks) than with architectural differences. This finding provides valuable guidance for protocol model selection and optimization in future implementations.
>
> This empirical evidence suggests that while capability gaps between models merit careful consideration, the framework's performance is more significantly influenced by alignment in training objectives than by architectural differences. These insights will inform both future optimizations of the framework and protocol model selection criteria.
>
> We thank the reviewer for highlighting these important considerations, which have helped us better articulate the relationship between protocol model design and framework performance.

---

> ### Author Response · Authors · 2024-12-01
>
> Thank you for your valuable feedback on our submission. We have thoroughly addressed all your comments and believe that our responses have reasonably resolved the concerns you raised. As the discussion period is coming to a close soon, we kindly ask if you could review our responses at your convenience. If you have any further questions or require additional clarification, please let us know--—we are more than willing to provide any additional information you might need.
>
> Regards,
> Authors of Submission3152

---

> ### Comment · Reviewer_m4Vi · 2024-12-01
>
> I really appreciate the authors’ efforts. Those experiment and analysis are valuable and did address some of my concern about the protocol selection procedure. Please make sure to include those discussion in the revised version. I would love to raise my score to 6.

---

> ### Author Response · Authors · 2024-12-03
>
> Thank you for your thorough feedback. Your comments have been invaluable in improving our paper.

---

### Official Review · Reviewer_XjTw · 2024-11-03

**Soundness:** 3
**Presentation:** 3
**Contribution:** 3
**Rating:** 8
**Confidence:** 4

**Summary:**

This work focuses on collaborative perception for heterogeneous agents and proposes a task- & model- agnostic framework that is scalable, efficient & secure. The framework involves learning a shared latent representation space, referred to as the protocol feature, using BEV features from LiDAR input. For each agent, a learned local adapter and reverter modules are used to map the agent-specific BEV features to the protocol feature space and vice versa. The protocol feature space is learned jointly across agents and tasks, while the local adapters and reverters are trained separately. Since the protocol feature space is learned once and adapter-reverter modules are lightweight, the proposed framework is scalable and efficient. Moreover, using a shared feature space prevents the need to share information about modalities & models from different agents, thus improving security. Extensive experiments on OPV2V and V2V4Real datasets show the effectiveness of the proposed framework in various scenarios.

**Strengths:**

- This work studies collaborative perception for heterogeneous agents and focuses on efficiency, scalability & security, which is important from a practical perspective.
- The shared latent representation space only needs to be learned once and the adapter-reverter modules are lightweight (~1MB), thereby reducing computational overhead (7.2x saving) and improving efficiency.
- Using a shared feature space avoids sharing information about the sensors, models & tasks, making it task-& model-agnostic and secure.
- The ideas are intuitive and the paper is well written & easy to follow.
- Experiments in both simulated (OPV2V) and real (V2V4Real) scenarios show the effectiveness of the proposed framework over several baselines in terms of detection performance (Tab.2,3), scalability (Fig.2), efficiency (Fig.2), better robustness to noise (Tab.1), being task-agnostic (Tab.4) & model-agnostic (Tab.4).
- Ablation study on different architectural components (Fig.3,4) and visualization of features & outputs (Fig.5) help to the capabilities of the proposed framework.

**Weaknesses:**

- The protocol feature space is learned using BEV features from LiDAR data. Is there a way to extend this to incorporate other modalities like RGB as well since dense semantic features from RGB complement sparse geometric features from LiDAR. It would also enhance the modality-agnostic aspect of the proposed framework and might scale better to real-world datasets.
- In the current experiments, the model- and task- agnostic setting is considered on the simulated OPV2V dataset. Is there any reason why this cannot be extended to real-world datasets like nuScenes? This would be helpful to verify if the trends hold on real-world datasets as well. This is not required for rebuttal but additional clarifications would be helpful.
- For multi-group collaborative systems, it seems like the agents might need to share extra information to form groups, e.g. which is the weaker modality. This might affect the modality agnostic or security aspects of the proposed framework. It'd be useful to provide some more insights into this.

**Questions:**

Some aspects need clarification, which are mentioned in the Weaknesses above.

---

> ### Author Response · Authors · 2024-11-27
>
> > ### Weakness 1: The protocol feature space is learned using BEV features from LiDAR data. Is there a way to extend this to incorporate other modalities like RGB as well since dense semantic features from RGB complement sparse geometric features from LiDAR. It would also enhance the modality-agnostic aspect of the proposed framework and might scale better to real-world datasets.
>
> We appreciate the reviewer's insightful suggestion regarding multi-modal protocol features. Our extensive experiments with different protocol model modalities confirm this intuition and reveal several important findings:
> - Modality Alignment Effect: Agents consistently perform better when paired with protocol models using similar input modalities, suggesting a natural affinity in feature space.
> - Multi-Modal Advantage: Most notably, a protocol model combining camera and LiDAR inputs improves performance across all agents, indicating successful feature complementarity. The multi-modal protocol model shows particular promise in enhancing performance even for single-modality agents.
>
> Here are our detailed experimental results:
> | Protocol Encoder Type   | Protocol Task          | Agent 1          | Agent 2          | Agent 3          | Agent 4          |
> |--------------------------|------------------------|-------------------|-------------------|-------------------|-------------------|
> | Non-Collab              | -                      | 0.941            | 0.399            | 0.548            | 0.675            |
> | Camera-based            | Object Det.           | 0.931 (−0.010)   | 0.777 (+0.368)   | 0.580 (+0.032)   | 0.671 (-0.004)   |
> | Camera + Lidar | Object Det.       | 0.937 (-0.004)   | 0.762 (+0.363)   | 0.632 (+0.084)   | 0.714 (+0.039)   |
>
> These findings suggest that incorporating multi-modal features in protocol models represents a promising direction for improving STAMP framework. We take this to be a guidance for further investigating protocol model design in future research.

---

> ### Author Response · Authors · 2024-11-27
>
> > ### Weakness 2: In the current experiments, the model- and task- agnostic setting is considered on the simulated OPV2V dataset. Is there any reason why this cannot be extended to real-world datasets like nuScenes? This would be helpful to verify if the trends hold on real-world datasets as well. This is not required for rebuttal but additional clarifications would be helpful.
>
> We strongly agree that real-world dataset evaluation is crucial for validating our approach, but the current real-world collaborative perception datasets present significant limitations. For instance, V2V4Real and DAIR-V2X provides only 3D bounding box labels, making it challenging to evaluate heterogeneous collaborative perception comprehensively.
>
> To ensure maximum experimental rigor within current dataset limitations, we employ two complementary approaches: simulating real-world conditions through Gaussian noise injection in the OPV2V dataset, and conducting 3D object detection experiments on the real-world V2V4Real dataset.

---

> ### Author Response · Authors · 2024-11-27
>
> > ### Weakness: For multi-group collaborative systems, it seems like the agents might need to share extra information to form groups, e.g. which is the weaker modality. This might affect the modality agnostic or security aspects of the proposed framework. It'd be useful to provide some more insights into this.
>
> In this paper, we did not explicitly propose or finalize how agents form groups in a collaborative system.  However, we have carefully considered the practical implications of multi-group collaboration and envision a secure, flexible system with the following characteristics:
>
> - Groups are formed through rigorous certification processes (e.g., V2V communication credentials) prior to the on road driving while agents must pass specific tests to receive group credentials
>
> - Agents only exchange information within trusted, credentialed groups so information sharing is protected by credential verification
>
> - Agents can hold multiple credentials simultaneously. Each agent can adapt their feature maps to multiple protocol presentations. This enables flexible participation across different collaborative groups
>
> For future development of practical deployment protocols, we plan to collaborate with transportation researchers and industry partners to design more comprehensive and realistic collaborative systems.
>
> We thank the reviewer for raising these important practical considerations and hope our response adequately addresses their concerns.

---

> > ### Comment · Reviewer_XjTw · 2024-11-29
> > **Thanks for all the clarifications**
> >
> > I appreciate all the clarifications provided by the authors. I'll retain my score.

---

> > > ### Author Response · Authors · 2024-12-03
> > >
> > > Thank you for your detailed reviews and constructive feedback. Your insights have greatly improved our paper.

---

### Official Review · Reviewer_fgQh · 2024-11-04

**Soundness:** 3
**Presentation:** 3
**Contribution:** 2
**Rating:** 8
**Confidence:** 5

**Summary:**

This paper introduces STAMP (Scalable Task- and Model-Agnostic Collaborative Perception), a framework designed to enable efficient, secure, and scalable multi-agent collaborative perception (CP) in autonomous driving systems. Recognizing the challenges posed by heterogeneous agents—such as varying sensors, models, and tasks—STAMP employs lightweight adapter-reverter pairs to align Bird’s Eye View (BEV) features to a unified protocol, allowing agents to collaborate without sharing model details. The framework is validated on simulated (OPV2V) and real-world (V2V4Real) datasets.

**Strengths:**

1. STAMP initiates the first study of task-agnostic and model-agnostic collaborative perception, and verifies the effectiveness of the proposed method in connected and autonomous driving scenarios.

2. STAMP introduces a unique adapter-reverter mechanism to bridge heterogeneity gaps in multi-agent collaboration.

2. STAMP addresses a critical need in autonomous driving by enabling heterogeneous agents to collaborate effectively, setting a foundation for more secure, scalable CP frameworks applicable to various downstream tasks.

**Weaknesses:**

1. While the proposed method claims scalability, the experiments include only three agents, which limits the demonstration of scalability in larger, more complex multi-agent systems.

2. Although the method is presented as task-agnostic, generalizing to untrained downstream tasks may prove challenging, suggesting that the adaptability across broader task domains could benefit from further validation.

3. The writing needs to be polished. The main paper contains numerous intricate details that might be better suited for the appendix, and the main figure contains redundant information that could be streamlined for clarity.

4. The framework focuses on vehicle-to-vehicle communication, which may narrow its broader impact within the ICLR community. Expanding the scope or exploring applications outside autonomous driving could increase its relevance.

5. The paper could benefit from a more comprehensive literature review, as some highly relevant works on efficient and scalable collaborative perception are missing [1-7]. Including a broader range of recent studies would provide a stronger context for the contributions and better situate the proposed framework within the current state of research.

[1] Li, Y., Ren, S., Wu, P., Chen, S., Feng, C. and Zhang, W., 2021. Learning distilled collaboration graph for multi-agent perception. Advances in Neural Information Processing Systems, 34, pp.29541-29552.

[2] Li, Y., Ma, D., An, Z., Wang, Z., Zhong, Y., Chen, S. and Feng, C., 2022. V2X-Sim: Multi-agent collaborative perception dataset and benchmark for autonomous driving. IEEE Robotics and Automation Letters, 7(4), pp.10914-10921.

[3] Hu, Y., Fang, S., Lei, Z., Zhong, Y. and Chen, S., 2022. Where2comm: Communication-efficient collaborative perception via spatial confidence maps. Advances in neural information processing systems, 35, pp.4874-4886.

[4] Huang, S., Zhang, J., Li, Y. and Feng, C., 2024. Actformer: Scalable collaborative perception via active queries. ICRA 2024.

[5] Yang, D., Yang, K., Wang, Y., Liu, J., Xu, Z., Yin, R., Zhai, P. and Zhang, L., 2024. How2comm: Communication-efficient and collaboration-pragmatic multi-agent perception. Advances in Neural Information Processing Systems, 36.

[6] Su, S., Li, Y., He, S., Han, S., Feng, C., Ding, C. and Miao, F., 2023, May. Uncertainty quantification of collaborative detection for self-driving. In 2023 IEEE International Conference on Robotics and Automation (ICRA) (pp. 5588-5594). IEEE.

[7] Su, S., Han, S., Li, Y., Zhang, Z., Feng, C., Ding, C. and Miao, F., 2024. Collaborative multi-object tracking with conformal uncertainty propagation. IEEE Robotics and Automation Letters.

**Questions:**

Could the authors illustrate task-agnostic collaborative perception more (especially the difference compared to the prior work [1])? As this prior work can be trained without knowing downstream tasks. However, the proposed framework in this paper seems to be trained on some specific tasks and is hard to generalize to novel downstream tasks. The authors are suggested to illustrate the limitations and setups clearly.

[1] Yiming Li, Juexiao Zhang, Dekun Ma, Yue Wang, and Chen Feng. Multi-robot scene completion: Towards task-agnostic collaborative perception. In Conference on Robot Learning, pp. 2062–2072. PMLR, 2023c. 3, 5

---

> ### Author Response · Authors · 2024-11-23
>
> Thank you for your detailed review and insightful comments. Please kindly see below for our responses to your comments:
>
> > ### Could the authors illustrate task-agnostic collaborative perception more (especially the difference compared to the prior work [1])? As this prior work can be trained without knowing downstream tasks. However, the proposed framework in this paper seems to be trained on some specific tasks and is hard to generalize to novel downstream tasks. The authors are suggested to illustrate the limitations and setups clearly.
>
> The key difference between Yiming Li et al. [1] and our proposed work lies in the stage of collaboration: Yiming Li et al. [1] focus on late fusion (or late collaboration), while our work focuses on intermediate fusion (or intermediate collaboration).
>
> We define the goal of task-agnostic collaborative perception to be enabling agents to collaborate effectively without limiting by or requiring knowledge of other agents’ downstream tasks.
> - In late fusion approaches, such as the one proposed by Yiming Li et al. [1], this is achieved by developing a general scene completion model and sharing the scene completion results with other agents.
> - For intermediate fusion, we achieve task-agnostic collaboration by sharing BEV features among agents with no task information needed, thereby avoiding the need for prior knowledge about other agent’s downstream tasks. (Some concerns are raised by other reviewers that the performance of our frameworks highly rely on the downstream task chosen for training the protocol model. We conducted some experiments and and attach the experimental results below)
>
>
> We believe each method has its unique advantages depending on the scenario:
> - **V2I Scenario (Advantageous for Yiming Li et al. [1]):** In a vehicle-to-infrastructure (V2I) setting, deploying a general scene completion model on the infrastructure, as proposed by Yiming Li et al. [1], is a straightforward and efficient solution. The infrastructure can share the completed scene results with other agents, supporting collaboration without requiring task-specific information. On the other hand, deploying our framework in such a scenario would involve additional steps, including training a protocol network and ensuring all agents have their adapter-reverter pairs.
>
> - **V2V Scenario (Advantageous for Our Framework):** In a vehicle-to-vehicle (V2V) scenario, where agents on the road are equipped with heterogeneous autonomous driving models with different downstream objectives, the approach proposed by Yiming Li et al. [1] becomes challenging. It is impractical to enforce all agents to deploy a scene completion model. In contrast, our framework enables collaboration by letting each agent train an adapter-reverter pair. These pairs align the agents’ BEV features with a central protocol representation, facilitating seamless collaboration regardless of task heterogeneity.
>
>
> Here we present experimental results comparing different protocol model designs, analyzing variations in encoder types and downstream tasks. Comparing the last two rows with our baseline STAMP model, we observe that the choice of downstream task for protocol model training significantly impacts the overall framework's performance. Specifically, agents tend to perform better when their task objectives align with those of the protocol model. Based on these findings, we propose that using a "task-agnostic model" as introduced by Li et al. [1] for the protocol model represents a promising direction for future research.
>
>
> | Protocol                | Encoder Type           | Protocol Task      | Agent 1 (lidar+obj.)          | Agent 2 (cam.+obj.)        | Agent 3   (lidar+static. seg.)          | Agent 4 ( lidar+dyn. seg.)         |
> |-------------------------|------------------------|--------------------|-------------------|-------------------|--------------|-------------|
> | Non-Collab             | -                      | -                  | 0.941            | 0.399            | 0.548            | 0.675            |
> | STAMP                  | CNN-based              | Object Det.        | 0.936 (−0.005)   | 0.760 (+0.362)   | 0.624 (+0.076)   | 0.690 (+0.014)   |
> |  STAMP (ablations) | CNN-based              | Dyn. Obj. Seg.  | 0.935 (−0.006)   | 0.743 (+0.344)   | 0.624 (+0.076)   | 0.723 (+0.048)   |
> |   STAMP (ablations) | CNN-based              | Static Obj. Seg.   | 0.747 (-0.194)   | 0.412 (+0.013)   | 0.681 (+0.133)   | 0.235 (-0.440)   |
>
>
> [1] Yiming Li, Juexiao Zhang, Dekun Ma, Yue Wang, and Chen Feng. Multi-robot scene completion: Towards task-agnostic collaborative perception. In Conference on Robot Learning, pp. 2062–2072. PMLR, 2023c. 3, 5

---

> ### Comment · Reviewer_fgQh · 2024-11-26
>
> Thank you for the detailed response. The prior work by Yiming Li et al. [1] also incorporates shared intermediate features and can be trained with self-supervision, suggesting that late or intermediate fusion may not be the key distinction. Therefore, my key concerns are: (1) What is the key difference between your work and the prior work? (2) The proposed framework in this paper seems to be trained on some specific tasks and is hard to generalize to novel downstream tasks. I am open to raising my score if the authors can clearly explain these differences, discuss the limitations of their method, and include the missing related works in the related work section.

---

> ### Author Response · Authors · 2024-11-26
>
> > ### Key Differences from Prior Work
>
> The key difference between our work and Li et al. [1] lies in the scope of heterogeneity we address. While Li et al.[1] focuses on handling heterogeneous tasks with homogeneous input modalities and model architectures, STAMP is designed to handle comprehensive agent heterogeneity across three dimensions:
> - Input modalities (e.g., LiDAR, camera)
> - Model architectures
> - Downstream tasks
>
> > ### Task Generalization and Training Requirements
>
> We would like to point out that the collaborative feature alignment (CFA) process (training the adapter and reverter pairs) requires task-specific training, which is where we address the novel downstream tasks generalization issue.
>
> However, we do acknowledge that a protocol network that is trained with a specific downstream task may not be generalized well enough for all novel downstream tasks. For example, comparing the last two rows with our baseline STAMP model, we observe that the choice of downstream task for protocol model training significantly impacts the overall framework's performance. Specifically, agents tend to perform better when their task objectives align with those of the protocol model. We take this as one of the limitations of this work.
>
> | Protocol                | Encoder Type           | Protocol Task      | Agent 1 (lidar+obj.)          | Agent 2 (cam.+obj.)        | Agent 3   (lidar+static. seg.)          | Agent 4 ( lidar+dyn. seg.)         |
> |-------------------------|------------------------|--------------------|-------------------|-------------------|--------------|-------------|
> | Non-Collab             | -                      | -                  | 0.941            | 0.399            | 0.548            | 0.675            |
> | STAMP                  | CNN-based              | Object Det.        | 0.936 (−0.005)   | 0.760 (+0.362)   | 0.624 (+0.076)   | 0.690 (+0.014)   |
> |  STAMP (ablations) | CNN-based              | Dyn. Obj. Seg.  | 0.935 (−0.006)   | 0.743 (+0.344)   | 0.624 (+0.076)   | 0.723 (+0.048)   |
> |   STAMP (ablations) | CNN-based              | Static Obj. Seg.   | 0.747 (-0.194)   | 0.412 (+0.013)   | 0.681 (+0.133)   | 0.235 (-0.440)   |
>
>
> > ### Limitations and Future Directions
>
> We acknowledge two primary limitations of our current approach:
>
> 1. As demonstrated in our experimental results above, the framework's performance is influenced by the protocol model's architecture and downstream tasks. While task-specific protocol models work, using task-agnostic models as proposed in [1] represents a promising direction for improving generalization and robustness.
>
> 2. Current implementation requires all agents to use collaborative models trained on collaborative perception datasets. Given the high cost of annotating such datasets compared to single-agent data, reducing this dependency represents an important area for future research. Developing methods to decouple or minimize reliance on collaborative datasets could significantly improve practical applicability.
>
> Our work makes a contribution as the first framework to simultaneously handle three fundamental types of agent heterogeneity: input modalities, model architectures, and downstream tasks. While we have identified several limitations in our current approach, these challenges present clear opportunities for future research directions that will further advance multi-agent collaborative perception.

---

> ### Author Response · Authors · 2024-11-26
>
> > ### While the proposed method claims scalability, the experiments include only three agents, which limits the demonstration of scalability in larger, more complex multi-agent systems.
>
> We appreciate the concern about scalability evaluation. While our experiments demonstrate the framework's effectiveness with up to four heterogeneous agents, we acknowledge this does not fully showcase the framework's potential scalability. This limitation stems primarily from the constraints of existing collaborative perception datasets, which contain a maximum of five agents per scene. Creating datasets with larger scale and higher realism will be crucial for future research in this field.
>
> Nevertheless, we have provided empirical evidence of our framework's scalability through efficiency analysis in Section 4.2, where we report the number of training parameters and estimated training time for collaborative feature alignment with up to 12 agents. These metrics demonstrate our method's computational efficiency and scalability advantages over existing approaches. We believe these quantitative results, even without direct performance measurements, provide strong support for our framework's scalability to larger multi-agent systems.
>
>
>
> > ### The writing needs to be polished. The main paper contains numerous intricate details that might be better suited for the appendix, and the main figure contains redundant information that could be streamlined for clarity.
>
> We appreciate your feedback on the paper's organization and presentation. In our revised version, we have streamlined the main paper to focus on key concepts and necessary technical details while moving intricate detailed discussions to the appendix. We have also refined the main figure to present information more concisely and clearly. These changes help readers better grasp the core ideas while maintaining access to comprehensive technical details for interested readers.
>
>
>
> > ### The paper could benefit from a more comprehensive literature review, as some highly relevant works on efficient and scalable collaborative perception are missing [1-7]. Including a broader range of recent studies would provide a stronger context for the contributions and better situate the proposed framework within the current state of research.
>
> We thank you for bringing these references to our attention. In our revised version, we have expanded the literature review to include recent works on efficient and scalable collaborative perception.

---

> > ### Comment · Reviewer_fgQh · 2024-11-27
> >
> > Thank authors for addressing my major concerns. I am happy to raise my score to 8.

---

> ### Author Response · Authors · 2024-12-03
>
> We sincerely appreciate the reviewers' thoughtful comments and valuable suggestions.

---

### Official Review · Reviewer_MyQk · 2024-11-04

**Soundness:** 2
**Presentation:** 3
**Contribution:** 2
**Rating:** 3
**Confidence:** 4

**Summary:**

This paper presents a framework for collaborative perception that is argued to be salable, tasks-independent and model agnostic, which is also argued to be capable of dealing with heterogeneity in the agents and enhancing flexibility and security.

**Strengths:**

+ Heterogeneity in collaborative agents, especially in robotics outside of collaborative driving, is an important problem.

+ The proposed method is straightforward, and the writing is easy to follow.

**Weaknesses:**

- Novelty is a key concern. The proposed method is based on BEV representations and performs intermediate fusion using the BEVs. Each component also uses or marginally extends existing methods, and the novelty of each component is not well justified. For example, Collaborative Feature Alignment (CFA) simply uses the BEV from each agent and projects it to a unified feature space.

- The paper argues that the proposed approach can address heterogeneity in the agents; however, the solution is simply based on the BEV representation that is assumed to be computed by each agent.

- Similarly, the claim that the proposed method is model-agnostic is also an overstatement, primarily due to its reliance on the BEV representation.

- Why is the proposed method secure? The method uses a set of neural networks, so how can we ensure that the use of these networks is secure?

- What is the coordinate frame of the BEV representation used by each vehicle? If the BEV is ego-centric for each vehicle, how can the correspondence of street objects between agents be found?

**Questions:**

See above.

---

> ### Author Response · Authors · 2024-11-27
>
> > ### Question 1: Novelty is a key concern. The proposed method is based on BEV representations and performs intermediate fusion using the BEVs. Each component also uses or marginally extends existing methods, and the novelty of each component is not well justified. For example, Collaborative Feature Alignment (CFA) simply uses the BEV from each agent and projects it to a unified feature space.
>
> Thank you for highlighting the importance of clearly articulating the novelty of our work. We acknowledge that utilizing BEV representations and intermediate fusion has been explored in prior research. However, the core novelty of our approach lies in the development of a scalable, task- and model-agnostic collaborative perception framework that, to the best of our knowledge, is the first to address all three aspects of agent heterogeneity simultaneously: (Heterogeneous Modalities, Heterogeneous Model Architectures and Parameters, and Heterogeneous Downstream Tasks.)
> For the Collaborative Feature Alignment (CFA) module, while projecting features into a unified space might appear straightforward, we believe **implementing this in a manner that supports heterogeneity in modalities, models, and tasks is both novel and a significant contribution to the field.**. Our experiments also demonstrate performance gains and computational efficiency over existing methods, especially as the number of heterogeneous agents increases.
>
>
> > ### Weakness 2: The paper argues that the proposed approach can address heterogeneity in the agents; however, the solution is simply based on the BEV representation that is assumed to be computed by each agent.
>
> > ### Weakness 3: Similarly, the claim that the proposed method is model-agnostic is also an overstatement, primarily due to its reliance on the BEV representation.
>
> We acknowledge that our framework relies on BEV representations computed by each agent. However, agents are free to generate these BEV features using any sensors, models, or processing methods. The BEV serves as a practical common ground for feature alignment but does not constrain the agents’ internal designs.
>
> Our method addresses heterogeneity by allowing agents to maintain their unique characteristics while collaborating effectively. We believe the reliance on BEV representations does not undermine the model-agnostic nature of our approach; instead, it facilitates feature alignment across diverse agents.

---

> ### Author Response · Authors · 2024-11-27
>
> > ### Weakness 4: Why is the proposed method secure? The method uses a set of neural networks, so how can we ensure that the use of these networks is secure?
>
> Thank you for bringing up the important topic of security in collaborative perception systems. Our framework enhances security by limiting the information shared among agents. Specifically: Agents do not share their neural network architectures, parameters, or input modalities. This reduces the risk of adversaries exploiting vulnerabilities inherent in shared models. By keeping model details private, we prevent potential attackers from performing white-box adversarial attacks, which require knowledge of the victim’s model.
>
> During collaboration, agents only exchange their BEV features and physical location information necessary for feature alignment and fusion. This minimal information sharing helps maintain the privacy and security of each agent’s internal systems.
>
> **Empirical Validation:**
>
> To substantiate our claims, we conducted supplementary adversarial attack experiments on the object detection task using the V2V4Real dataset. Following the methodology of Tu et al. (2021) [1], we compared three settings:
> - End-to-End Training: Agents share full model parameters, enabling direct white-box attacks.
> - HEAL: Agents share encoders but have different fusion modules and decoders, with limited access to victim models.
> - STAMP (ours): Agents share only protocol feature representations, with no access to other agents’ models.
> Results:
>
> | AP@50          | End-to-end | HEAL  | STAMP (ours) |
> |-----------------|-------------|-------|--------------|
> | Before Attack   | 0.513       | 0.515 | 0.523        |
> | After Attack    | 0.087       | 0.506 | 0.503        |
>
> The results indicate that our method is robust against adversarial attacks, showing minimal performance degradation compared to the significant impact observed in the end-to-end training scenario. This empirical evidence supports our assertion that the proposed method enhances security by safeguarding agents against malicious attacks.
>
> [1] Tu, J., Wang, T., Wang, J., Manivasagam, S., Ren, M., & Urtasun, R. (2021). Adversarial attacks on multi-agent communication. In Proceedings of the IEEE/CVF International Conference on Computer Vision (pp. 7768-7777).

---

> ### Author Response · Authors · 2024-11-27
>
> > ### Weakness 5: What is the coordinate frame of the BEV representation used by each vehicle? If the BEV is ego-centric for each vehicle, how can the correspondence of street objects between agents be found?
>
> In our framework, each agent computes its BEV representation in its ego-centric coordinate frame. To enable accurate correspondence and fusion of street objects between agents, we employ the following approach:
>
> - Sharing of Location Information: Along with the BEV features, agents share their precise pose information, which includes their position and orientation in a global or common reference frame.
>
> - Coordinate Transformation: Upon receiving another agent’s BEV features and pose information, each agent performs a coordinate transformation to align the incoming features with its own coordinate frame. This ensures that the features correspond accurately to the same physical space.
>
> These two methods are standard practices in the field of multi-agent collaborative perception, which is why they were not detailed in the paper. However, we recognize that additional explanation could benefit readers who are less familiar with these techniques. We will clarify these methods in the revised manuscript to improve understanding for all readers.
>
> Besides, we recognize that in real-world scenarios, there may be errors in localization. To evaluate the robustness of our method, we conducted experiments where we introduced Gaussian noise to the agents’ pose information. The results, detailed in Section 4.2 of the paper, demonstrate that our framework maintains robust performance even in the presence of localization inaccuracies.
>
> By incorporating pose sharing and coordinate transformations, our method effectively aligns the BEV features from different agents, facilitating accurate correspondence of street objects and enhancing the overall collaborative perception

---

> ### Author Response · Authors · 2024-11-28
>
> We hope that our responses have effectively addressed your concerns and have clarified the contributions and novelty of our work. We are committed to refining our paper based on your valuable feedback. We would be grateful if you could reconsider your evaluation in light of these clarifications. Thank you again for your thoughtful review, which has helped us improve the quality and clarity of our manuscript.

---

> ### Author Response · Authors · 2024-12-01
>
> We sincerely appreciate your taking the time to review our manuscript and providing valuable feedback. We wanted to follow up to see if our previous responses have sufficiently addressed your concerns and clarified the unique contributions and novelty of our work. We are fully committed to refining our paper based on your valuable feedback. If you have any additional comments or concerns, please let us know---your thoughtful review has been instrumental in improving the quality and clarity of our manuscript, and we would be more than happy to address any remaining questions you may have.
>
> Regards,
> Authors of Submission3152

---

> ### Author Response · Authors · 2024-12-03
>
> We wanted to send a gentle follow-up regarding our submission. We greatly value your expertise and the time you've invested in reviewing our work. If you've had a chance to review our responses, we would greatly appreciate any additional feedback. If there are any remaining concerns we haven't fully addressed, we would be happy to provide further clarification.
>
> Regards,
>
> Authors of Submission3152

---

### Author Response · Authors · 2024-12-02
**Response to Reviewers: Clarifying STAMP's Novelty and Addressed Concerns**

## **Dear Reviewers,**

We sincerely thank you for your thoughtful and detailed feedback on our manuscript. We would like to address major concerns about the novelty of our work and summarize the revisions we made to address the reviewers' concerns.

> ### **Novelty and Contributions**

Our work makes several novel contributions to collaborative perception (CP):

**First Heterogeneity Framework**: While previous works have addressed individual aspects of agent heterogeneity, STAMP is the first framework to **simultaneously handle all three dimensions**: input modalities, model architectures, and downstream tasks.

**Lightweight, Scalable Design**: Our adapter-reverter mechanism provides an efficient solution for feature alignment, requiring only ~1MB additional parameters per agent. This represents a **7.2x reduction** in computational overhead compared to existing methods [1], while maintaining or even improving performance.

**Enhanced Security**: Our protocol framework inherently enhances security by eliminating the need to share agent details with other agents. The experiments show that STAMP maintains 98% performance under adversarial attack where end-to-end approaches degrade to 17% performance.

Regarding the novelty concern, we would like to emphasize that while the model architectures we used in this work are not the key novelty contribution, the heterogeneous collaborative framework design and the training and inference pipelines represent significant novel contributions. We believe this approach opens new possibilities for **scalable and secure heterogeneous collaborative perception system research**.

---

> ### **Our revisions**

**Protocol Model Selection**: In this revised version, we further clarify the architecture design of the protocol model. Our additional experiments with protocol models of different input modalities, encoder architectures, and training downstream tasks reveal that encoder architecture has minimal impact on framework performance, while input modalities and training objectives have relatively major effects. This discovery provides valuable guidance for future researchers in designing better protocol models.

**Adversarial Robustness**: We have included complementary experiments on the adversarial robustness of our proposed framework, demonstrating that STAMP enhances adversarial robustness by eliminating access to ego agent's information from other agents.

**Experimental Comparison with Existing Methods**: We conducted new experiments comparing STAMP with V2X-ViT, CoBEVT, and HMViT under heterogeneous input modality scenarios. The results show that STAMP not only supports all three types of agent heterogeneity (while other methods only support heterogeneous input modality) but also surpasses them in performance.

**Paper Writing Enhancement**: Following reviewer suggestions, we have: 1. Included additional highly related works, 2. Streamlined the main paper to focus on key concepts while moving detailed discussions to the appendix, and 3. Refined the main figure for a more concise and clear information presentation.

**Additional Performance Analysis**: Our training analysis comparing single-agent and end-to-end training reveals why STAMP outperforms end-to-end training strategies. We discovered that training multiple agents together in an end-to-end manner can cause overfitting or underfitting of some models, resulting in suboptimal checkpoint selection. This problem is effectively avoided by training each agent separately.

Since the PDF editing window closed before the completion of certain experimental results, we are committed to including these valuable complementary experiments and detailed analyses in the next version of our manuscript.

---

> ### **Future Research Opportunities**

**Investigating Protocol Model Selection**: Our experiments reveal that protocol model design significantly impacts framework performance, particularly regarding input modalities and training objectives. A promising direction is exploring input or task generic protocol models that potentially improve the framework's adaptability and robustness.

**Reducing Reliance on Collaborative Datasets**: STAMP currently requires each agent to be trained with collaborative datasets. Reducing this dependency could significantly improve training efficiency and cost-effectiveness.

**Enhancing Communication Efficiency**: While STAMP adapts to various BEV feature dimensions, incorporating more efficient communication strategies such as selective sharing warrants further investigation. We consider this a promising direction for future research.

We sincerely thank all reviewers for their valuable suggestions regarding complementary experiments and paper improvements, as well as for inspiring these promising future research directions.


Regards,

Authors of Submission3152

---

> ### **Reference**

[1] Lu et al. (2024). An extensible framework for open heterogeneous collaborative perception. ICLR.

---

### Meta-Review · Area_Chair_2MRT · 2024-12-20

**Metareview:**

This paper addresses multi-agent collaborative perception, a setting where multiple agents exchange perception-related information to improve sensing capabilities. Specifically, this paper assumes heterogeneous agents (that may be equipped with different sensors or models performing different perception tasks) and follows a mid-level fusion approach that fosters the exchange of mid-level information exchange and fusion.
The paper proposes BEV representation for feature exchange among agents and a shared communication protocol for feature exchange and fusion. Key findings (experiments on synthetic and real data) show the advantages of collaborative perception, with graceful degradation to baseline performance (single system).

The paper received ratings of 3, 6, 6, 8, 8. Overall, reviewers are on the positive side (avg. rating 6.2). Two endorsements are strong, two positive, and one reviewer argues against acceptance.

Reviewers agree that the problem is important and interesting (especially in the context of autonomous driving), the paper is very well written and structured, recognize that this paper is the first to study task-agnostic *and* model-agnostic collaborative perception, find that the effectiveness of the proposed contributions is clearly demonstrated in the context of AV data, comment that this work is an important foundation for future efforts in this domain, appreciate proposed systems efficiency and scalability, appreciate thorough ablation on the different architectural components, appreciate fantastic feature visualizations (before&after fusion).

Reviewers provided constructive feedback and engaged in a discussion with authors, after which all provided a positive rating, with the exception of reviewer MyQk. A detailed summary of this discussion is below- the key takeaway is that the reviewer did justify their claims regarding the lack of novelty or responded to author clarifications (and ACs reminders to provide a response and final ratings). I agree with the reviewer's remarks that the "heterogeneous" claim made in the intro is misleading. However, I agree with the author's justification and urge the authors to clarify this in the revised version of this paper.

AC specifically appreciates well-structured coverage of related work that portrays a broad coverage of collaborative perception. This makes this paper's contribution accessible to an unfamiliar audience with multi-agent perception. Overall, the presentation clarity is exceptional. Discussion on limitations and failure cases strengthens this work further.

**Additional Comments On Reviewer Discussion:**

All reviewers engaged in the discussion except for the reviewer MyQk.
The reviewer expressed concerns regarding novelty (the reviewer states that each component of the proposed work is an extension of the prior art but does not elaborate on which methods the reviewer had in mind). The reviewer also remarks that the paper emphasizes heterogeneity, while BEV representation is assumed to be computed individually by each agent. Similarly, the reviewer finds the claim that this approach is model-agnostic and an overstatement (the approach relies on BEV).

The authors provided a detailed response- I agree with the authors' comment that the claim is supported by the fact that "agents are free to generate these BEV features using any sensors, models, or processing methods" (however, I understand where reviewer's objection originates from- after reading the intro, I also assumed that this method would work with *any* model, which is not the case, this should be clarified).

Without a detailed explanation of why this paper lacks novelty and with respect to which methods, I am discarding the objection regarding novelty. The reviewer had an opportunity to clarify their statement but didn't.

The reviewer did not respond to the author's clarifications and AC's request to comment on the rebuttal and provide their final rating. Based on this, AC finds that the reviewer does not make a strong case for rejecting this paper.

---

### Decision · Program_Chairs · 2025-01-22

Accept (Poster)